# Oxidative stress-driven enhanced iron production and scavenging through Ferroportin reorientation worsens anemia in antimony-resistant *Leishmania donovani* infection

Souradeepa Ghosh[1], Krishna Vamshi Chigicherla[1], Shirin Dasgupta[2], Yasuyuki Goto[3], Budhaditya Mukherjee[1]*

1 School of Medical Science and Technology, Indian Institute of Technology, Kharagpur, West Bengal, India, 2 Dr B C Roy Multispeciality Medical Research Centre, Indian Institute of Technology, Kharagpur, West Bengal, India, 3 Laboratory of Molecular Immunology, Graduate School of Agricultural and Life Sciences, The University of Tokyo, Tokyo, Japan

* bmukherjee@smst.iitkgp.ac.in, aditya26884@gmail.com

## Abstract

Despite the withdrawal of pentavalent-antimonials in treating Visceral leishmaniasis from India, recent clinical isolates of *Leishmania donovani* (LD) exhibit unresponsiveness towards pentavalent-antimony (LD-R). This antimony-unresponsiveness points towards a genetic adaptation that underpins LD-R's evolutionary persistence and dominance over sensitive counterparts (LD-S). This study highlights how LD evolutionarily tackled antimony exposure and gained increased potential of scavenging host-iron within its parasitophorous vacuoles (PV) to support its aggressive proliferation. Even though anti-leishmanial activity of pentavalent antimonials relies on triggering oxidative outburst, LD-R exhibits a surprising strategy of promoting reactive oxygen species (ROS) generation in infected macrophages. An inherent metabolic shift from glycolysis to Pentose Phosphate shunt allows LD-R to withstand elevated ROS by sustaining heightened levels of NADPH. Elevated ROS levels on the other hand trigger excess iron production, and LD-R capitalizes on this surplus iron by selectively reshuffling macrophage-surface iron exporter, Ferroportin, around its PV thereby gaining a survival edge as a heme-auxotroph. Higher iron utilization by LD-R leads to subsequent iron insufficiency, compensated by increased erythrophagocytosis through the breakdown of SIRPα-CD47 surveillance, orchestrated by a complex interplay of two proteases, Furin and ADAM10. Understanding these mechanisms is crucial for managing LD-R-infections and their associated complications like severe anemia, and may also provide valuable mechanistic insights into understanding drug unresponsiveness developed in other intracellular pathogens that rely on host iron.

**Data availability statement:** RNA-Sequencing was performed utilizing external service by Bionivid Project (GSE279792, https://www.ncbi.nlm.nih.gov/geo/query/acc.cgi?acc=GSE279792). All other relevant data are within the manuscript and its Supporting information files.

**Funding:** This work is supported by grants from ICMR (6/9-7(297)/2022-ECD-II) and STARS (MoE-STARS/STARS-2/2023-0516) to BM and grants JSPS 22H05057, 24K02271 to YG. The funders had no role in study design, data collection and analysis, decision to publish, or preparation of the manuscript.

**Competing interests:** The authors have declared that no competing interests exist.

## Author summary

The highlight of this study is on understanding the mechanisms by which *Leishmania donovani,* unresponsive to antimony (LD-R), has adapted to prolonged exposure to antimonials. This adaptation resulted in LD-R evolving into a persistent strain, which causes organ parasite overload and severe anemia during clinical infection. The common mode-of-action of both pentavalent-antimonials and host-defense-arsenal includes Reactive-oxygen-species (ROS)-outburst, which LD-R successfully exploits for their survival benefit. Our study portrays that while LD-R manages to endure heightened ROS levels through a metabolic shift in central-carbon metabolism, high ROS paradoxically benefits LD-R proliferation by providing access to excess iron for these heme-auxotrophs. LD-R has devised a strategy to rapidly propel host-iron inside its parasitophorous vacuole (PV) through re-orientation of macrophage-surface iron exporter-Ferroportin around the PV membrane. The enhanced utilization of iron sustains LD-R's aggressive proliferation and differentiation into amastigotes, causing iron deficiency that triggers SIRPα cleavage. The breakdown of SIRPα results in the loss of the discriminatory signal between CD47-enriched live RBCs and CD47-deficient senescent RBCs, thereby promoting inflated erythrophagocytosis. We identified two crucial host-proteases, Furin and ADAM10, acting concurrently to cleave SIRPα at the surface of LD-R-infected macrophages, contributing to the mechanistic understanding of the severe anemia observed in clinical LD-R infection.

## Introduction

*Leishmania donovani* (LD), is an intracellular protozoan parasite causing fatal visceral leishmaniasis (VL). The mainstay for VL, pentavalent antimonials (SbV), was discontinued in treating Indian sub-populations for decades due to the emergence of resistance. Despite this, recent clinical isolates still show unresponsiveness to SbV, indicating stable genetic adaptations [1,2]. Studies have demonstrated that patients infected with antimony-resistant LD (LD-R) tend to have a parasite overload in their spleen and liver compared to those infected with sensitive strains (LD-S) [3]. Successful LD-R survival and proliferation inside mammalian hosts, to prevail as persistent strain, involves evading host-defenses and resisting anti-leishmanial effects of antimony, both of which trigger reactive oxygen species (ROS) to eliminate LD. ROS functions as a double-edged sword in shaping the outcome of LD-infection: on one hand, it plays a vital role in eradicating the pathogen, on the other hand, a controlled level of ROS is necessary for intracellular amastigote differentiation [4]. The major contributor of ROS is the multi-subunit complex: NADPH oxidase (NOX) comprising six subunits, three of which (p47, p40, and p67) undergo phosphorylation and assemble around the phagolysosomal-membrane harboring intracellular-LD [5]. To date, most of the intracellular pathogens, including *Leishmania*, have been reported to delay or impair NOX assembly and evade oxidative stress [6-8]. Therefore, it is appealing to investigate whether LD-R also delays ROS production or adapts to combat ROS-outbursts to support its rapid intracellular proliferation. Interestingly, ROS has been intricately linked to iron, an essential nutrient required for the proliferation and pathogenicity of *Leishmania* [9]. However, Kinetoplastida including *Leishmania*, have lost the ability to synthesize heme. Although, *Leishmania* has partially regained the final three enzymes of the heme-biosynthesis pathway through horizontal gene transfer [10], primarily they are heme-auxotrophs reliant on host-iron for their survival. Thus, it can be hypothesized that LD-R should scavenge host-iron within their parasitophorous

vacuole (PV) more efficiently than LD-S to support their aggravated intracellular proliferation. However, little is known about the molecular events that might contribute to increased iron-acquisition while preventing iron-efflux from LD-R-infected-macrophages (MΦs). This work, deciphers the mechanisms underlying iron-acquisition/utilization, by focusing on a range of iron-metabolizing/transporters that may facilitate rapid proliferation of LD-R outcompeting LD-S. MΦs also play a key role in iron homeostasis by recycling senescent RBCs through erythrophagocytosis which might explain the mechanism of anemia in response to LD-infection, which remained elusive. Importantly, higher acquisition of host-iron can also explain the frequent incidence of Coombs-positive hemolytic anemia among VL-positive cases from the Indian subcontinent [11]. This study identifies molecular events leading to severe anemia in response to LD-R infection and also throws light into the mechanism of the rapid spread of antimony-unresponsiveness in VL through intricate utilization of host resources, thereby conferring selective survival advantages to LD-R.

## Results

### 1. Antimony-resistant *Leishmania donovani* (LD-R) outcompete drug-sensitive isolates (LD-S) in an experimental model of murine-infection

Previous studies suggested that LD-R-infection consistently leads to higher organ-parasite burden in both mice and human VL-patients [3,12]. Initially, increased metacyclogenesis in LD-R was thought to be the primary factor contributing to higher *in-vivo* infectivity, resulting in organ-parasite overload [3,12,13]. However, later findings indicated that LD-R promastigotes exhibit greater replicative potential compared to LD-S counterparts, potentially providing a selective intracellular survival advantage as compared to LD-S [14]. An equal number of metacyclic LD-S-GFP (AG83) and LD-R-RFP (BHU575) promastigotes were sorted and competitively infected in murine peritoneal macrophages (MΦs) (Fig 1A.i, 1A.ii, and 1A.iii). Live-cell imaging and video microscopy revealed that while equal numbers of LD-S-GFP and LD-R-RFP promastigotes entered host-MΦs initially, LD-R-RFP outcompeted LD-S-GFP amastigotes, indicating increased intracellular proliferation and survival (Fig 1A.iv and S1 Video). To determine whether this increased survival fitness is not restricted to only BHU575, MΦs were again infected with an equal number of sorted metacyclics for two clinical LD-R (BHU575, BHU814) and LD-S (AG83, BHU777) isolates (S1A Fig), which also revealed no significant difference in initial infectivity but a significantly higher number of intracellular amastigotes for LD-R isolates 24 hrs pi with same trend being followed at 48hrs and 72hrs pi (S1B.i, S1B.ii and S1B.iii Fig). Hence, BHU575 was selected as representative LD-R and AG83 as representative LD-S to conduct all further comparisons with the inclusion of other representative strains as and when mentioned. Further, *in-vivo* experiments in BALB/c mice infected with an equal number of sorted metacyclic, i.e., either with LD-S-GFP (Group:1), with LD-R-RFP (Group:2), with a mixture of LD-S-GFP: LD-R-RFP in 50:50 ratio (Group:3), or with LD-S-GFP: LD-R in 80:20 ratio (Group:4) corroborated these findings. Firstly, a significantly larger infected spleen with increased weight was observed in Group:2 (~120.566mg), Group:3 (~118.266mg), and Group:4 (~116.633mg) mice as compared to Group:1 (~32.766mg) mice spleen (Fig 1B.ii) which was used for calculating LDU. In Groups: 2, 3, and 4, the infected-spleen LDU index was significantly higher than in Group:1 (Fig 1C.i), indicating a higher infection-load compared to Group:1. This observation was in concurrence with amastin expression showing the highest splenic amastigote load for Group:2-infected-mice, followed by Group:3 and Group:4, all of which are significantly higher compared to Group:1-infected-mice (Fig 1C.ii). Fluorescence microscopy of the splenic sections (Fig 1C.ii) revealed RFP-positive LD-R amastigotes outcompeted

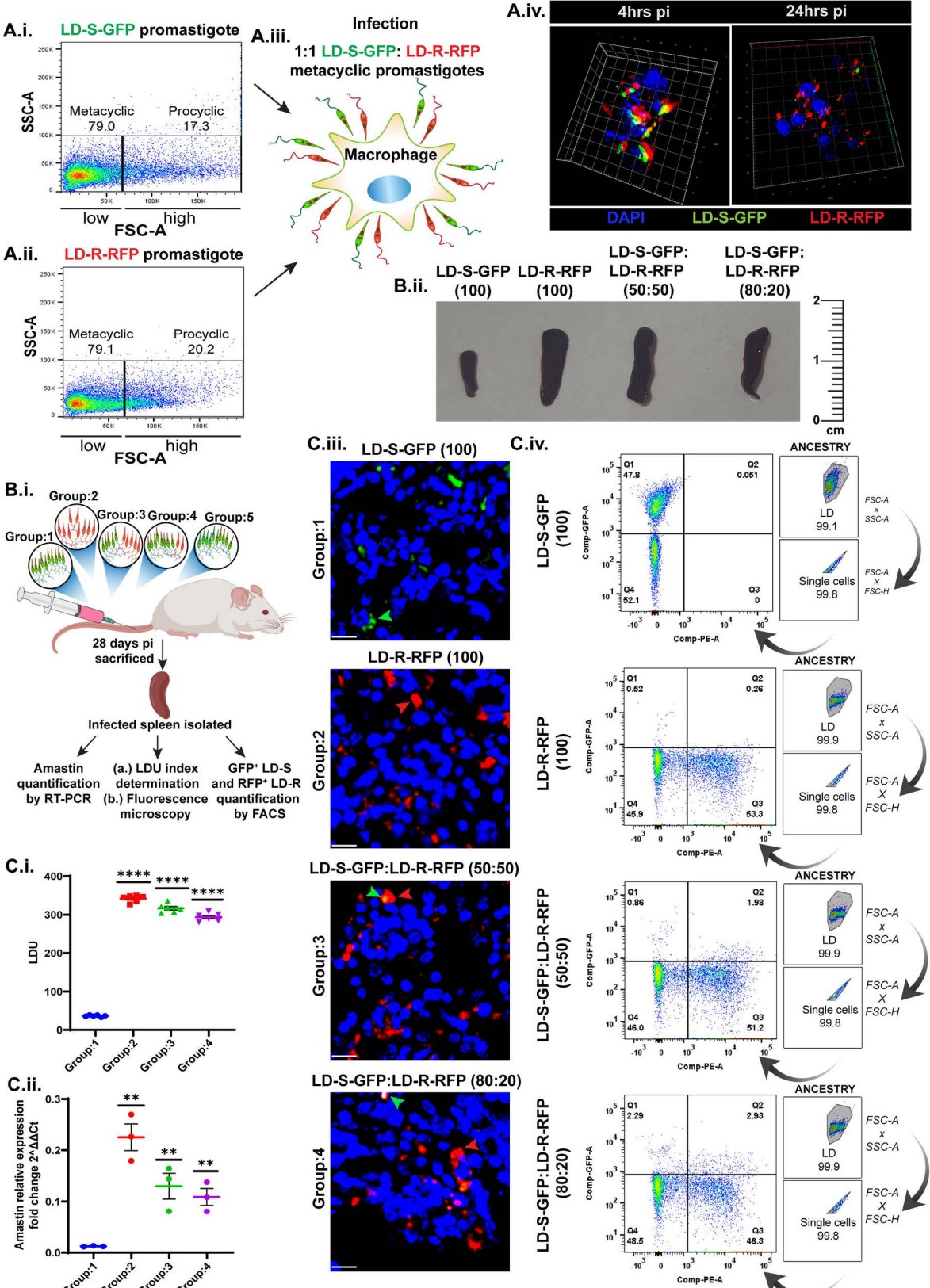

**Fig 1. LD-R outcompetes LD-S both in *in-vitro* and *in-vivo* infection:** Representative flow cytometer images (BD LSRFortessa Cell Analyzer) showing the % of metacyclics and procyclic LD-promastigote population in the culture of (A.i.) GFP-expressing

antimony-sensitive (LD-S-GFP) and **(A.ii.)** RFP-expressing antimony-resistant (LD-R-RFP) after uniform sorting in Beckman Coulter Cytoflex srt. FSC$^{low}$ (left) represents the metacyclic population and FSC$^{high}$ (right) represents the procyclic population. **(A.iii.)** Scheme representing an equal number of LD-S-GFP and LD-R-RFP metacyclic promastigotes infection in peritoneal murine macrophages (MΦs). **(A.iv.)** 3-D Z-stack rendering of Super-resolution image of mixed infection of LD-S-GFP and LD-R-RFP parasites in MΦs at 4 hrs pi (left panel) and 24 hrs pi (right panel). **(B.i.)** Scheme showing tail-vein infection of BALB/c mice with LD-S-GFP (Group:1), or with LD-R-RFP (Group:2), or with LD-S-GFP: LD-R-RFP in 50:50 (Group:3), or with LD-S-GFP: LD-R-RFP in 80:20 (Group:4), or with LD-S-GFP: LD-S in 50:50 (Group:5). 28 days pi mice were sacrificed and the spleen was isolated from each experimental mouse, weighed and cut into 3 pieces for i. Amastin quantification by qRT-PCR; ii. (a.) LDU index determination from the macerated spleen by Giemsa staining. (b.) Fluorescence microscopy of cryosectioned spleen; and iii. GFP$^+$ LD-S and RFP$^+$ passage 1 promastigote population enumeration by flow cytometry. **(B.ii.)** Representative spleen image of infected-spleen 28 days pi from mice representing Groups: 1–4 (from left to right) with a scale bar (cm) at the right for differential comparison. **(C.i.)** Graph showing LDU in each experimental set (Groups: 1–4). N = 3 for each group in duplicate. **(C.ii.)** Relative amastin expression fold change (2^ΔΔCt) in the infected-spleen isolated 28 days pi from Groups:1–4 (N = 3 for each group) was determined from qRT-PCR depicting the amastigote load in each set with respect to murine β-actin. **(C.iii.)** Confocal images of the cryosectioned spleen sample 28 days pi for experimental groups: 1–4 representing the LD-S-GFP and LD-R-RFP amastigote load (green dots marked in green arrow and red dots marked in red arrow) in each splenic sample. Scale bars indicate 20 μm. **(C.iv.)** % of the GFP-positive population enumerated from flow cytometry to denote the load of LD-S-GFP in comparison to LD-R in experimental Groups: 1–4. The right-most panel shows the ancestry of each analysis. Each experiment was performed in triplicates and graphs are represented as Mean with SEM. **P** ≤ 0.0001 is marked as ****, and **P** ≤ 0.01 is marked as **.

GFP-positive LD-S amastigotes around the splenic MΦs in Group:3 and Group:4 infected spleen. Group:1 (LD-S-GFP) and Group:2 (LD-R-RFP) infected spleen were used as infected control where also it was evident that LD-R-RFP shows higher amastigote load as compared to LD-S-GFP (Fig 1C.iii). Furthermore, to confirm that the loss of GFP-signal in splenic MΦs for Group:3 and Group:4 is not due to plasmid loss, another mixed murine-infection of LD-S-GFP and untagged-LD-S (50:50) was performed (Group:5), which resulted in significant retention of GFP-signal among splenic amastigotes (S1C.i and S1C.ii Fig). Finally, flow cytometry-based quantification of the individual group (Fig 1C.iv), revealed ~47.8% of the pure GFP-positive population in the case of Group:1, which gets significantly reduced to ~0.86% for Group:3, 2.29% in Group:4. Parallelly, pure RFP-positive population for Group:2 was ~53.3% which was maintained at 51.2% and 46.3% for Group:3 and Group:4 further confirming the notion that LD-R has an inherent increased potential for *in-vivo* proliferation surpassing its sensitive counterparts. Notably, it took approximately 10–13 days for Group:1, 5–7 days for Group:2, and 6–8 days for Group:3 and Group:4, for respective LD-promastigotes to emerge from the infected macerated spleens. Susceptibility testing against SbV showed similar EC$_{50}$ values for independent LD-lines emerging from Group:3 and Group:4 mice (~20.01 and ~20.00 respectively) as compared to the parental LD-R-line (EC$_{50}$ ~21.04) (S1D Fig and Table 1), further supporting the notion that increased intracellular survival of LD-R contributes to organ-parasite overload.

## 2. LD-R-infection rather than suppressing host-induced ROS promotes its induction

To investigate if LD-R's enhanced proliferation is linked with suppression of intracellular ROS generation, ROS levels were quantified in LD-S and LD-R-infected-MΦs at early time points: 2hrs, 4hrs, and 8hrs pi keeping uninfected MΦs as control. However, LD-R-infected-MΦs exhibited higher ROS levels at 2hrs and 4hrs pi compared to LD-S-infected-MΦs, which flattened out by 8hrs pi (Figs 2A.i, 2A.ii and S1E). However, incubation of MΦs with killed-LD-R failed to generate substantial ROS, indicating that replicating LD-R is required for ROS generation (S1E Fig). It is well established that LD prevents NOX assembly in the phagolysosomal membrane to delay ROS production [8]. However, significant co-localization of p47 (NOX subunit) with Rab5a (early endosomal marker) was noticed surrounding LD-R-PV in infected-MΦs at 4hrs pi (Fig 2B.i and 2B.ii), which was significantly

Table 1. Table showing $EC_{50}$ value against SbV as determined from the dose-response curve (S1D Fig).

| Samples | LD-S-GFP | LD-R | LD-S-GFP: LD-R (50:50) | LD-S-GFP: LD-R (80:20) |
|---|---|---|---|---|
| $EC_{50}$ | 1.833 | 21.04 | 20.01 | 20.00 |
| $EC_{50}$ range | 1.552 to 2.141 | 18.84 to 23.40 | 18.60 to 21.50 | 17.33 to 22.92 |

less for LD-S-PV as reported previously [15]. This trait of early ROS induction seems to be specifically linked with primary antimony-unresponsiveness as infection with lab-generated Amphotericin-B-unresponsive LD-isolates (AmpB-R-DD8) without primary unresponsiveness towards antimony failed to elicit ROS production in infected-MΦs (S1F Fig). In contrast, LD-R strains with unresponsiveness only towards antimony (D10), or towards multiple other drugs but with primary unresponsiveness towards antimony (BHU575) [2,16] could successfully induce ROS production (S1F Fig). This ability of LD-R to withstand high ROS levels was attributed to its high intrinsic level of NADPH. HPLC-based quantification of NADPH/ $NADP^+$ in LD-S and LD-R suggests that the NADPH/$NADP^+$ ratio is significantly higher in LD-R as compared to LD-S at a retention time of around 1.5–2mins. (Fig 2C and 2C inset), which neutralizes intracellular ROS [17]. HPLC-based quantification of NADPH/$NADP^+$ in LD-S and LD-R suggests that the NADPH/$NADP^+$ ratio is significantly higher in LD-R as compared to LD-S at a retention time of around 1.5–2mins. MΦs infected with NADPH exhausted LD (NADPH$^{exh}$-LD) [18] (mentioned in Methods), resulted in significantly lower intracellular amastigote burden as compared to untreated LD-R infection, confirming the role of NADPH in neutralizing ROS without affecting initial infectivity (Fig 2D.i and 2D.ii). Contrarily, for NADPH$^{exh}$-LD-S-infection, a slight increase in parasite burden was observed in 24 hrs as compared to 4hrs pi (Fig 2D.i and 2D.ii). Measurement of ROS in MΦs infected with NADPH$^{exh}$-LD-R revealed a continuous persistence of ROS similar to LD-S-infection even after 8hrs pi (S1G Fig).

This observation also suggested that LD-R as compared to LD-S might be preferentially utilizing PPP which acts as the major source of NADPH. Gene-Set Enrichment Analysis (GSEA) of GSE144659 comparing clinical LD-R-isolates (BHU575, BHU814) and LD-S-isolates (AG83, BHU777) [16] was performed to further confirm this. GSEA for glycolysis showed down-regulation in LD-R-isolates as compared to LD-S-isolates with an enrichment score of –0.29430503, and normalized enrichment score of –1.2197618 with specific subset genes for the Glycolysis pathway contributing towards downregulation (Figs 2E.i and S1H and Table A in S1 Table). While for PPP, GSEA revealed an upregulation for both the LD-R-isolates, with an enrichment score of 0.3644994 and a normalized enrichment score of 1.577721, with specific subset genes of the PPP contributing towards the upregulation (Fig 2E.ii and Table B in S1 Table). Furthermore, the downregulation of Glycolysis and upregulation of the PPP pathway in LD-R-isolates is also reflected by a low FDR value (0.043337647), suggesting an inherent metabolic shift in LD-R towards PPP.

Targeted metabolomics from extracted metabolites of Central carbon metabolism by LC-MS confirmed this metabolic shift towards PPP for LD-R. While a 3D-plot representing the coverage and total metabolite extraction for both LD-R and LD-S (Fig 2F.i) showed an efficient extraction, annotation, and quantification revealed a significant enrichment of NADPH, trypanothione, and acetyl CoA for LD-R, specifically for LD-R (Fig 2F.ii). No significant change for glyceraldehyde-3-phosphate and a significant decrease in $NADP^+$ level in LD-R compared to LD-S (Fig 2F.ii) further confirm an inherent metabolic switch in clinical LD-R-isolates towards PPP.

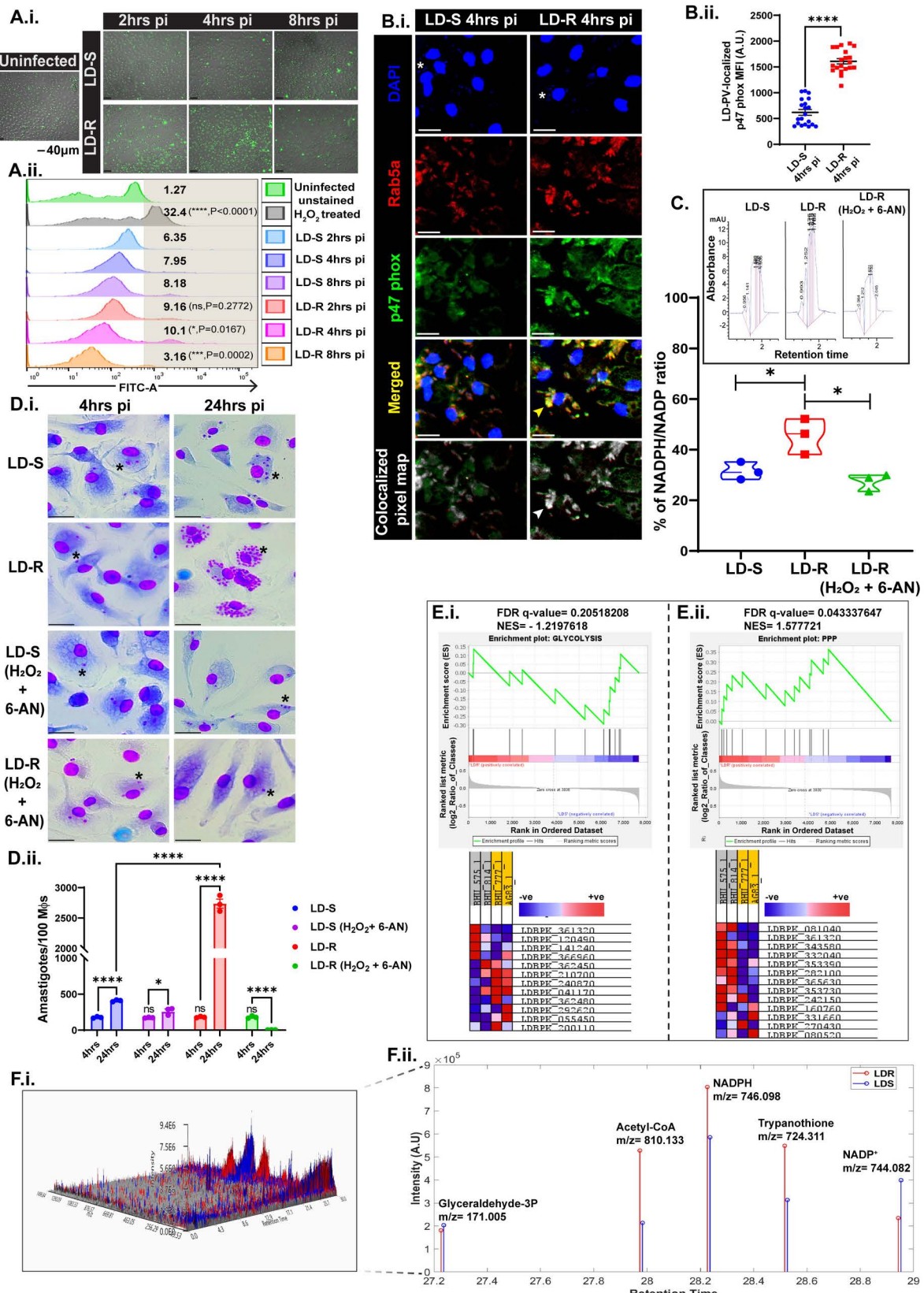

**Fig 2. An inherent metabolic shift in LD-R helps these parasites neutralize host-generated ROS outbursts. (A.i.)** 20X-magnification live-cell confocal images of H₂DCFDA-stained (green) uninfected MΦs, LD-S, and LD-R infected-MΦs at early hours (2, 4, and 8 hrs pi).

Scale bar indicates 40μm. **(A.ii.)** Representative half offset histogram plot representing live-cell flow cytometry quantification of DCF fluorescence in FITC-A filter in LD-S and LD-R infected MΦs at 2 hrs, 4 hrs, and 8 hrs pi. Uninfected unstained control was used for gating the macrophage population and 100 μM $H_2O_2$ treated macrophage positive control (1 hr treatment) was used to gate the DCF-positive population in the FITC-A channel. The grey area demarcates the FITC-A-positive population. $H_2O_2$-treated macrophages showed significant ROS outbursts as compared to uninfected unstained control (****, **P** ≤ 0.0001). LD-R 2hrs pi showed no significant change as compared to LD-S 2hrs pi (ns, **P** > 0.05), while at 4 hrs pi, LD-R showed significant ROS outburst compared to LD-S infection (*, **P** ≤ 0.05), and at 8 hrs pi LD-R shows a significant drop in ROS level as compared to LD-S infection (***, **P** ≤ 0.001), N = 3 independent biological repeats. **(B.i.)** Confocal images showing the expression pattern of p47 phox (green) around early PV demarcated by Rab5a (Red) in LD-S and LD-R infected-MΦs at 4 hrs pi. The colocalized pixel map (right-most panel) shows the colocalized region of p47 phox and Rab5a marked in white, the intensity of which is directly proportional to the percentage of colocalization. **(B.ii.)** Dot-plot showing the Mean Fluorescence Intensity (MFI) of LD-harboring-PV localized with p47 phox in infected MΦs (N = 20). **(C.)** Graph representing % of NADPH/NADP turnover in LD-S, LD-R, or NADPH$^{exh}$-LD-R (by $H_2O_2$ + 6-AN treatment) metacyclic promastigotes quantified in HPLC using NADPH and NADP-standards, with inset plot representing the HPLC-chromatogram of NADPH. **(D.i.)** Giemsa-stained images of LD-S, LD-R, NADPH$^{exh}$-LD-S, and NADPH$^{exh}$-LD-R-infected-MΦs at 4 hrs and 24 hrs pi. **(D.ii.)** Bar graph representing Amastigotes/100 MΦs of experimental sets of D.i. **(E.)** GSE analysis showing glycolysis **(E.i.)** and PPP **(E.ii.)** enrichment plot in LD-R as compared to LD-S. The lower panel shows heat maps expression pattern of **i.** glycolysis (left) and **ii.** PPP (right) genes. Red and blue signify positive and negative enrichment respectively. **(F.i.)** A 3D snapshot of the whole metabolome showcasing all the resolved and MS data. **(F.ii.)** Stem-plot showing metabolites having specific m/z values with retention time on the X-axis and their intensity on the Y-axis. Red denotes LD-R and blue denotes LD-S metabolites. All other scale bars except A.i. indicate 20μm. One representative small nucleus of LD has been marked in (**\***) to show the infected-MΦs in Fig 2B and 2E.i. Each experiment was performed in triplicates, and graphs are represented as Mean with SEM. **P** > 0.05 is marked as 'ns' (non-significant), **P** ≤ 0.05 is marked as *, and **P** ≤ 0.0001 is marked as ****.

## 3. ROS acts as a beneficial tool to produce labile iron in LD-R-infected-macrophages

Since a significantly higher level of initial ROS in LD-R-infected-MΦs was observed, its contribution in the higher proliferative potential of LD-R was investigated as opposed to its widely accepted role in eliminating LD-infection [19]. LD-S and LD-R infection were performed in MΦs in the presence of a suboptimal dose of antimony ~1.52 μg/ml (equivalent to $EC_{50}$ value for LD-S, AG83) [16]. While this suboptimal dose of antimony resulted in a significant decrease of intracellular amastigotes for LD-S as anticipated, it resulted in a significant increase in the number of intracellular LD-R-amastigotes at 24 hrs pi as compared to untreated control (Fig 3A.i, **top 2$^{nd}$ panel and** 3A.ii.). However, the presence of antioxidant N-acetyl-L-cysteine (NAC), resulted in a significant decrease in intracellular LD-R-amastigotes, as opposed to LD-S (Fig 3A.i, **3$^{rd}$ panel and** 3A.ii.). Similar infection with NADPH$^{exh}$-LD-R, in the presence of suboptimal SbV resulted in a drastic reduction in LD-R-amastigote count suggesting that inherent NADPH of LD-R helps these parasites to thrive and proliferate in high ROS (Fig 3A.i, **bottom panel and** 3A.ii.). Taken together, these results imply that ROS might be beneficial for intracellular LD-R proliferation as opposed to LD-S-infection. An earlier study on *Trypanosoma* infection in mice similarly found that oxidative stress contributes to the parasite's ability to persist [20], by promoting the degradation of ferritin, which stores iron as $Fe^{3+}$, thereby facilitating the release of bioavailable $Fe^{2+}$ [21]. $Fe^{2+}$ is a vital nutrient for LD replication, and LD being a heme auxotroph, relies upon the host for its acquisition. Comparison of Ferritin levels between LD-S and LD-R infected-MΦs at 4hrs pi showed no major change in Ferritin (FTH-1) level (Fig 3B.i and 3B.ii.a), suggesting $Fe^{2+}$ mobilized from Ferritin might not be the source of higher iron to support aggressive replication of LD-R. Apart from Ferritin, another source of $Fe^{2+}$ can be provided by Heme oxygenase-1 (HO-1), which breaks down heme into iron ($Fe^{2+}$) [10,22]. As opposed to Ferritin, a consistently increased HO-1 expression was observed in LD-R as compared to LD-S-infected-MΦs at 4hrs pi (Fig 3B.i and 3B.ii.b). Interestingly, *in silico* analysis with Eukaryotic promoter database and PROMO revealed p50 (−1284/−1277, −635/−624, and −483/−476) and c-Rel (−1230/−1211, −913/−904, −633/−624, and −603/−594) binding site in

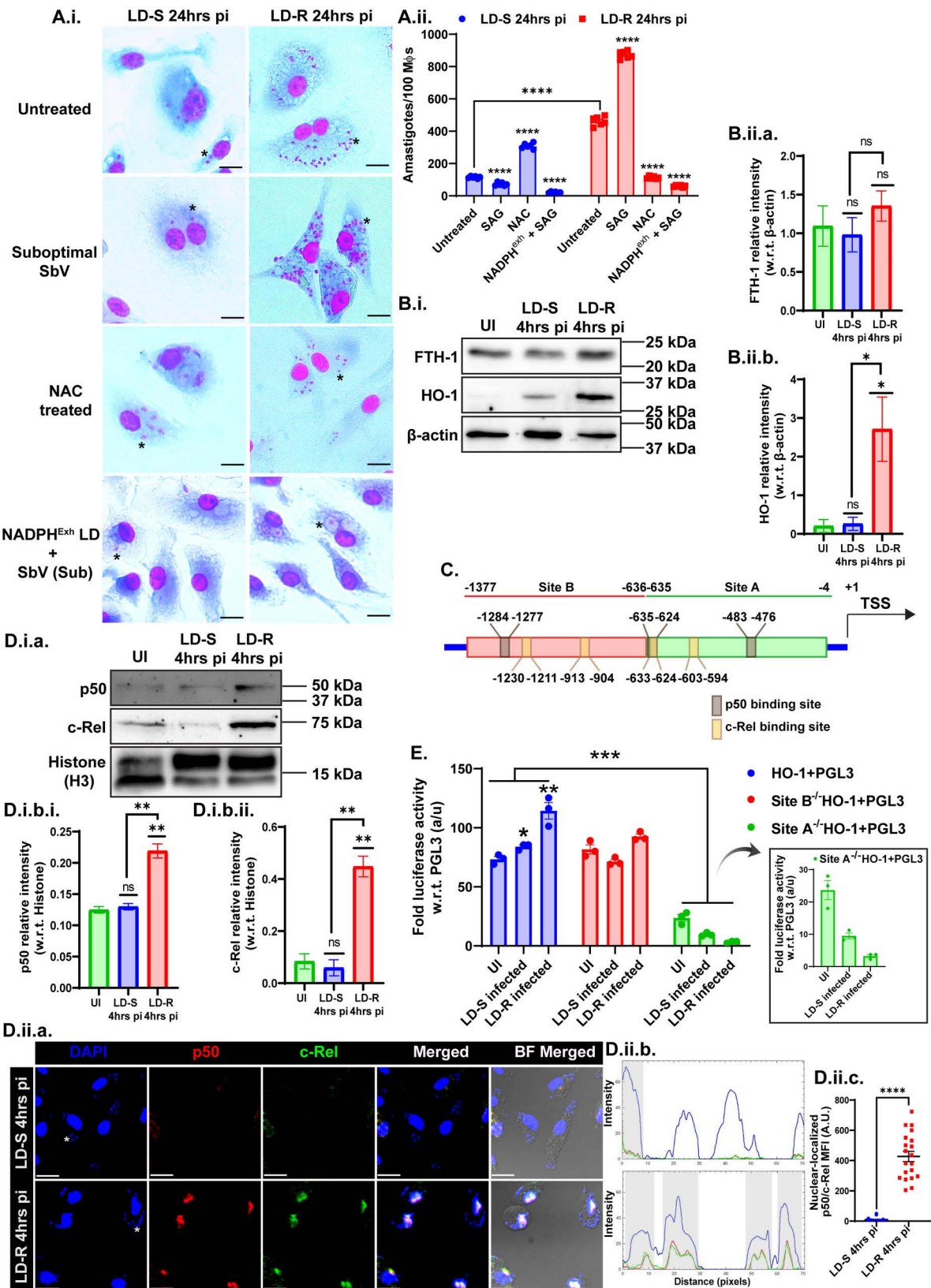

**Fig 3. LD-R proliferates more in high ROS by upregulating HO-1 activity. (A.i.)** Giemsa-stained images of LD-S, LD-R, in murine peritoneal MΦs with or without the presence of a suboptimal dose of SbV (1.52 µg/ml). In some experimental conditions LD-infection was

performed in the presence of N-acetyl-L-cysteine (NAC), while in some experimental conditions, LDs are pre-treated with $H_2O_2$ + 6-AN. All the infections were performed for 24 hrs **(A.ii.)** Bar graph showing amastigotes/100MΦs in all the experimental sets mentioned in A.i. **(B.i.)** Western blot of whole cell lysate showing significant high expression of Heme-oxygenase 1 (HO-1) in LD-R-infected-MΦs while no significant change in expression level was observed for Ferritin (FTH-1) at 4 hrs pi. β-actin is used as the positive control. **(B.ii.a. and B.ii.b.)** Relative intensity analysis of FTH-1 and HO-1, i.e., fold change with respect to β-actin for whole cell lysate. **(B.ii.a.)** No significant change in FTH-1 was observed in any experimental conditions (ns, **P** > 0.05). **(B.ii.b.)** Significant upregulation of relative intensity of HO-1 is observed in LD-R 4 hrs pi as compared to both UI and LD-S 4hrs pi (*, **P** ≤ 0.05) whereas no significant change is observed in LD-S 4hrs pi as compared to UI control (ns, **P** > 0.05). **(C.)** Schematic representation of HO-1 promoter region with p50 and c-Rel binding sites. Site A (−4/−635) was demarcated as green, and Site B (−636/−1377) was demarcated as red. **(D.i.a.)** Western blot of nuclear fraction showing significant high expression of p50 and c-Rel in LD-R-infected-MΦs 4 hrs pi with Histone (H3) as loading control. **(D.i.b.i. and D.i.b.ii.)** Relative intensity analysis of p50 and c-Rel, i.e., fold change with respect to Histone for nuclear fraction. Significant enrichment of nuclear p50 and c-Rel respectively observed in LD-R 4 hrs pi as compared to UI and LD-S 4hrs pi (P ≤ 0.01 marked as **) whereas LD-S 4hrs pi shows no significant change (ns) compared to UI (P > 0.05). Each densitometry analysis is presented as a bar graph of Mean ± SEM for 3 biological replicates. **(D.ii.a.)** Confocal images representing the localization of p50 and c-Rel, in MΦs, with DAPI representing the nucleus. One representative small nucleus of LD has been marked in (*) to show the infected-MΦs. **(D.ii.b.)** RGB-profile plot with grey regions demarcating the region where p50 and c-Rel are colocalized with DAPI. **(D.ii.c.)** Dot-plot showing Mean Fluorescence Intensity (MFI) of nuclear-localized p50/c-Rel for 20 LD-S or LD-R-infected-MΦs. **(E.)** Fold luciferase activity of RAW264.7 cell lysate transfected either with HO-1+PGL3 promoter, or Site A (−/−), Site B(−/−) deleted constructs followed by infection with LD-S and LD-R for 4 hrs. PGL3 enhancer empty vector is used for normalization for each transfected set. Each experiment was performed in 3 biological replicates and graphical data are represented as Mean with SEM. **P** ≤ 0.05 is marked as *, **P** ≤ 0.01 is marked as **, **P** ≤ 0.001 is marked as ***, and **P** ≤ 0.0001 is marked as ****.

the promoter region of *hmox-1* (Fig 3C). It has been previously reported that LD-R-infection and not LD-S, results in specific activation of p50/c-Rel-dependent transcriptional activation in infected-MΦs [23]. Western blot analysis, coupled with confocal microscopy showed a significant co-translocation of p50/c-Rel in the nucleus of LD-R-infected-MΦs as early as 4hrs pi corroborating with the activation of HO-1 (Fig 3D.i.a, 3D.i.b.i, 3D.i.b.ii, 3D.ii.a, 3D.ii.b, 3D.ii.c, 3B.i and 3B.ii.b, respectively). Finally, Luciferase assay with wild-type, or truncated promoter constructs: Site A−/− (−4/−635) and Site B−/− (−636/−1377) in the presence of LD-R and LD-S infection, further confirmed specific interaction of p50/c-Rel with Site A of HO-1 promoter resulting in its specific upregulating in LD-R-infected MΦs (Fig 3E). Interestingly, some luciferase activity modulation in HO-1 promoter construct in response to LD-S-infection as compared to uninfected control was observed, probably owing to the effect of some other transcription factors having binding sites within HO-1 promoter.

## 4. LD-R scavenges labile iron in their PV through the reshuffling of host Ferroportin

Higher HO-1 expression in LD-R-infected-MΦs at 4hrs pi (Fig 3B.i and 3B.ii.b), should result in increased labile iron pool ($Fe^{2+}$) as compared to LD-S-infected-MΦs. However, live-cell imaging and plate-reader-based quantification of Calcein-AM fluorescence [24–26] revealed diminished labile iron pool in LD-R-infected-MΦs (Fig 4A.i, 4A.ii and 4A.iii). To track $Fe^{2+}$ infiltration into LD-R-PV, lysotracker red coupled with Calcein-AM and sucrose density gradient-based ferrozine quantification [27] were employed. Live-cell imaging showed rapid iron infiltration into LD-R-PV at 4hrs pi, which was confirmed by Ferrozine-based quantification (Figs 4B, S2A.i and S2A.ii). Interestingly, for LD-R-infected-MΦs, a slight increase in $Fe^{2+}$ level in the cytoplasm was noted 24 hrs pi along with a significant drop in LD-R-PV as compared to 4hrs pi probably indicating a rapid utilization of iron by LD-R. To determine the key players that could drive the rapid mobilization of the labile iron into LD-R-PV as opposed to LD-S-PV, several well-reported iron regulatory/ metabolizing and transporter proteins were screened by performing a comparative RNAseq analysis between LD-S and LD-R-infected-MΦs 4hrs and 24 hrs pi with uninfected-MΦs as reference.

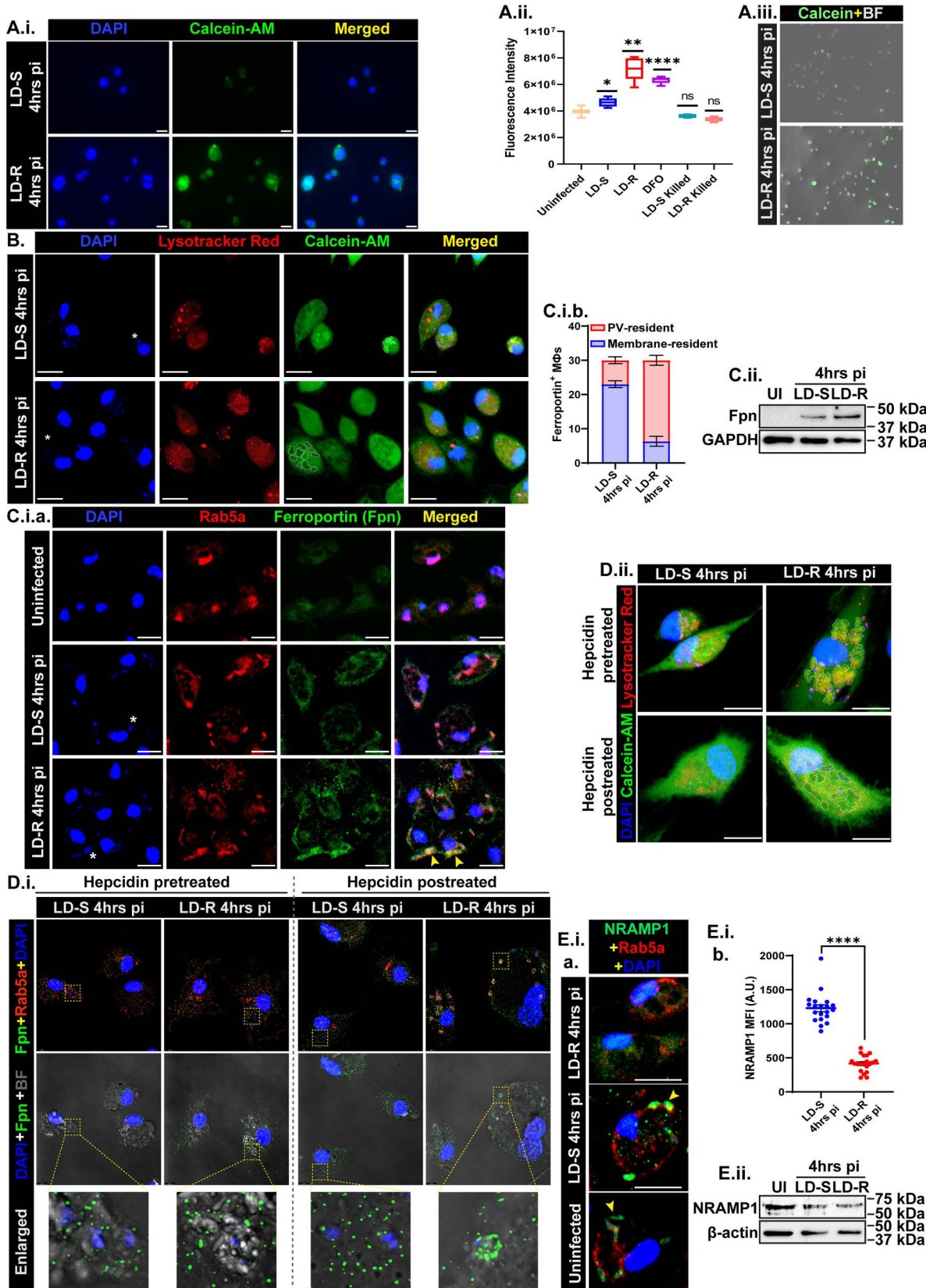

**Fig 4. Reshuffling of Ferroportin and NRAMP1 around LD-R-PV acts as a strategy to rapidly scavenge iron. (A.i.)** Live-cell Calcein-AM-stained images of LD-S (upper panel) and LD-R-infected-MΦs (lower panel) at 4 hrs pi. **(A.ii.)** Box-plot showing fluorescence

intensity of Calcein in MΦs either uninfected or LD-S or LD-R-infected, or 100 µM DFO-treated, or killed LD-S, or LD-R-incubated at 4 hrs as Mean with SEM. (**A.iii.**) 10X live-cell images of Calcein-AM-stained (green) LD-S and LD-R infected-MΦs at 4 hrs pi. (**B.**) Live-cell images of Calcein-AM and Lysotracker Red (labels the PV)-stained MΦs infected with LD-S and LD-R at 4 hrs pi with PV marked in dotted areas (3rd panel). (**C.i.a.**) Confocal images showing Ferroportin (Fpn) localization by labeling with anti-Ferroportin antibody (green) and anti-Rab5a antibody (Red). The yellow arrow signifies the colocalization of Ferroportin with Rab5a and DAPI. (**C.i.b.**) Stacked bar-graphs showing the number of PV-resident and membrane-resident Ferroportin among 30 LD-S and LD-R-infected MΦs in triplicates. (**C.ii.**) Western blot showing Ferroportin expression with GAPDH as the loading control. (**D.i.**) Super-resolution images of MΦs either pretreated with hepcidin followed by LD-infection (left 2 panels) or LD-infection followed by hepcidin posttreatment (right 2 panels). The uppermost panel shows the merged channel of Fpn (Green), Rab5a (Red), and DAPI (Blue). The middle panel shows the merged channel of Fpn and DAPI with Brightfield (BF). The lowermost panel shows the enlarged view of Fpn localization with regard to intracellular LD (small blue dots representing LD nucleus). (**D.ii.**) Live-cell images showing iron levels in similar experimental sets as D.i. staining with Calcein-AM and Lysotracker Red. The dotted region shows the PV portion with distinct changes. (**E.i.a.**) Confocal images showing NRAMP1 expression with Rab5a (Red) and DAPI (blue). (**E.i.b.**) Dot-plot showing Mean Fluorescence Intensity (MFI) of NRAMP1 expressed around PV-residing intracellular LD (N = 20). (**E.ii.**) Western blot showing NRAMP1 expression keeping β-actin as the loading control. One representative small nucleus of LD has been marked in (**\***) to show the infected-MΦs. Scale bars indicate 20µm. Each experiment was performed in triplicates. **P** > 0.05 is marked as 'ns' (non-significant), **P** ≤ 0.05 is marked as **\***, **P** ≤ 0.01 is marked as **\*\***, and **P** ≤ 0.0001 is marked as **\*\*\*\***.

Transcriptomics analysis showed a significant change in a large number of transcripts in response to LD-S or LD-R infection (S2B.i Fig). At 4hrs pi, both LD-S and LD-R infection showed an inclination towards M1 phenotype, with LD-R showing a higher degree of M1 polarization state possibly due to higher ROS induction (S2B.ii Fig and S1 Sheet). However, at later time points, LD-R infections showed a complete shift towards the M2 phenotype. This huge shift towards the M2 phenotype might help LD-R to survive and proliferate more inside the host as compared to LD-S which showed relatively less inclination towards the M2 phenotype (S2B.ii Fig and S1 Sheet). Although transcriptomics analysis showed a significant change in a large number of transcripts in response to LD-S or LD-R infection (S2B.i Fig) however, in concurrence with a previous report [28], no significant change in the transcript expression of iron metabolizing or transporters like *trf* (CD71, Transferrin receptor 1), *slc40a1* (Ferroportin), *slc11a1* (NRAMP1), *slc11a2* (DMT1) were observed between LD-S and LD-R-infected-MΦs (S2C Fig).

Previous reports have suggested that LD-infection can regulate the expression of host iron-regulatory factors by post-transcriptional/post-translational modifications which have been reported for CD71, Ferroportin, and NRAMP1 [27,29-31]. Co-localization and Western blot analysis of CD71 showed significantly increased uniform expression around both LD-S and LD-R-PV (S3A.i, S3A.ii.a and S3A.ii.b Fig), suggesting a basal level of iron might be provided by CD71 for both LD-S and LD-R as reported earlier [29]. Interestingly, for Ferroportin (sole iron exporter on MΦ surface), a significant amount was found to be colocalized around early PV harboring LD-R as opposed to LD-S (Fig 4C.i.a and 4C.i.b and S2 Video). Moreover, both LD-S and LD-R-infection resulted in a significant increase in total Ferroportin expression at 4hrs pi as compared to uninfected control, although this increase appears to be slightly more in LD-R-infected-MΦs (Figs 4C.ii and S3B). To prove that this selective re-localization of Ferroportin might drive iron mobilization inside LD-R-PV, MΦs were pre-treated with Hepcidin (1µg/ml), which degrades Ferroportin when exposed to MΦs-surface [32] and then infected with LD-S or LD-R. Super-resolution microscopy of both Hepcidin pre-treated and post-treated conditions for LD-S-infected-MΦs showed no significant accumulation of Ferroportin around LD-S (Figs 4D.i and S3C). However, for LD-R-infection, while no Ferroportin was observed around intracellular LD-R in Hepcidin pre-treatment, post-treatment of Hepcidin resulted in a significant accumulation of Ferroportin around LD-R, clearly suggesting that LD-R-infection reshuffles Ferroportin from the host plasma membrane to its PV rendering Hepcidin-mediated degradation of Ferroportin ineffective (Fig 4D.i, **rightmost**

**panel**). Simultaneous live-cell imaging with Calcein-AM and lysotracker Red revealed the disappearance of ferroportin in Hepcidin pre-treated MΦs fails to pump iron inside LD-R-PV as observed by higher calcein fluorescence similar to LD-S-infection (Figs 4D.ii, **upper right panel and S3D, left panel**). Contrarily, hepcidin post-treatment results in a rapid reshuffling of ferroportin around the PV membrane with no impact on iron trafficking inside LD-R-PV, while Hepcidin pre- and post-treatment has no bearing on LD-S-PV (Fig 4D.ii, **left panel**). Notably, NRAMP1 which has been previously implicated in the process of iron export from *L. major*-PV [27], also reflected no significant change in transcript level between LD-S and LD-R infection (S2C Fig). However, immunofluorescence showed reduced NRAMP1 presence around the early LD-R-PV compared to LD-S-PV, although no significant change in total NRAMP1 level was observed between LD-S and LD-R-infected-MΦs in Western blot, suggesting that NRAMP1 might be more selectively excluded from LD-R-PV (Figs 4E.i.a, 4E.i.b, 4E.ii, S3E and S3F). These results altogether suggested that both the iron-scavenging and iron-withholding capacity of LD-R-PV is much more than LD-S-PV and is mainly regulated by post-transcriptional reprogramming of iron-regulators in host MΦs.

## 5. Iron depletion in the cytoplasm of LD-R-infected-MΦs quench excess ROS through non-canonical activation of NRF2 at early hours

Excessive trafficking of iron within LD-R-PV leads to a decline in cytoplasmic iron levels in LD-R-infected-MΦs. Previous studies have suggested that a decrease in cytoplasmic iron triggers the activation of p62/SQSTM1, which subsequently degrades the NRF2 inhibitor Keap1 [33]. Consequently, NRF2 translocates into the nucleus and binds to the antioxidant response element (ARE)-promoter, initiating the antioxidative pathway. Thus, iron deprivation appears to be the primary mechanism responsible for reducing elevated ROS in LD-R-infected-MΦs at a later time point (Fig 2A.i and 2A.ii). Significantly increased expression of p62 in the cytoplasm of LD-R-infected-MΦs similar to DFO-treated-MΦs 4hrs pi further enforced this observation (S4A.i.a, A.i.b and A.ii Fig). Elevated p62 level is also consistent with increased NRF2 expression in the whole cell lysate and nuclear fraction of LD-R-infected-MΦs compared to LD-S-infected-MΦs or uninfected-MΦs at the same time point (S4B.i.a and S4B.i.b Fig). Furthermore, confocal microscopy with RGB-profile plot coupled with super-resolution 3-D rendering Z-stack images, also revealed a significant translocation of NRF2 in the nucleus in response to LD-R-infection, while NRF2 remains predominantly in the cytoplasm of LD-S-infected-MΦs (S4B.ii.a, B.ii.b and S4B.iii Fig). These observations indicate LD-R-infection results in reduced iron levels in the host cytoplasm around 4hrs pi leading to p62-mediated activation of NRF2, which subsequently reduces ROS levels at a later time point.

## 6. Iron insufficiency in later time points is compensated by inflated erythrophagocytosis driven by SIRPα degradation

Previously, the quantification of iron in the cytoplasmic and PV-fraction of LD-R-infected-MΦs using a Ferrozine-based colorimetric method revealed a slight yet significant increase in cytoplasmic iron levels 24 hrs pi (S2A.ii Fig). This suggested that a drop in cytoplasmic iron levels prompts LD-R-infected-MΦs to uptake more iron from the medium to sustain their rapid proliferation. Conventionally, the recycling of senescent phagocytosed RBCs is the primary source of iron for MΦs [34]. Comparison of erythrophagocytosis between LD-R and LD-S infected-MΦs at 4hrs and 24 hrs pi, showed, no significant difference in RBC uptake initially. However, at 24 hrs pi, LD-R-infected-MΦs exhibited a significant accumulation of ingested RBCs, notably co-localizing with GFP-positive LD (Fig 5A.i and 5A.ii.). Fluorometric

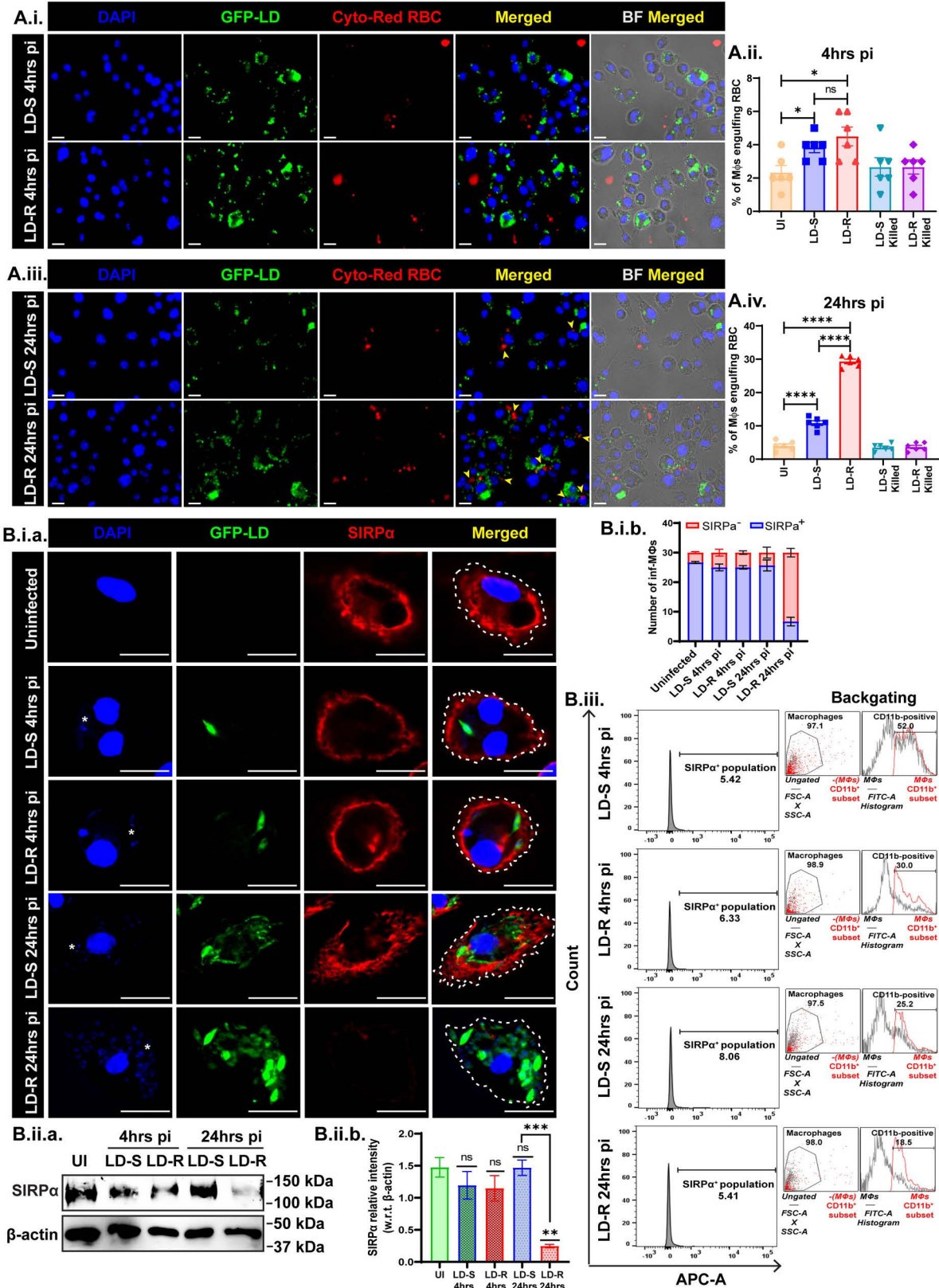

**Fig 5. Augmented erythrophagocytosis in LD-R-infected-MΦs at 24 hrs pi in concomitance with SIRP α degradation.(A.i.)** Confocal images show no significant erythrophagocytosed RBCs in both LD-S and LD-R infected-MΦs at 4 hrs pi. DAPI stains nucleus, LD

were labeled with GFP (green), and RBCs were labeled with Cyto-Red dye after infection to check engulfed RBCs in LD-infected-MΦs. **(A.ii.)** Bar graph showing % of MΦs engulfing RBCs calculated from different fields of independent experimental triplicates at 4 hrs pi. Killed-parasite-infected-MΦs were kept in control. **(A.iii.)** Confocal images showing significant engulfed RBCs in LD-R-infected-MΦs at 24 hrs pi. The yellow arrow indicates ingested RBCs in infected-MΦs. **(A.iv.)** Bar graph showing % of MΦs engulfing RBCs calculated from different fields of experimental triplicates sets at 24 hrs pi. **(B.i.a.)** Confocal images showing SIRPα expression (red) at the macrophage surface outline demarcated by a dotted line in infected-MΦs with LD expressing GFP (green). One representative small nucleus of LD has been marked in (*) to show the infected-MΦs. **(B.i.b.)** Stacked bar graph showing the number of SIRPα+ and SIRPα- infected MΦs among 30 MΦs represented as SEM for 3 independent experiments. **(B.ii.a.)** Western blot of whole cell lysate showing SIRPα expression in uninfected-MΦs, LD-S, LD-R infected-MΦs at 4 hrs pi and 24 hrs pi. β-actin is used as a housekeeping control. **(B.ii.b.)** Relative intensity analysis of SIRPα, i.e., fold change with respect to β-actin for whole cell lysate. Significant downregulation of relative intensity of SIRPα in LD-R 24 hrs pi as compared to UI (**, $P \leq 0.01$) and LD-S 24hrs pi (***, $P \leq 0.001$). The densitometry analysis is presented as a bar graph of Mean ± SEM for 3 biological replicates. **(B.iii.)** Flow-cytometric data showing histogram plot representing SIRPα expression in CD11b-positive MΦ population analyzed in FlowJo v10. The right-most panel shows the back-gating of individual experimental sets. Scale bars indicate 20 μm. Each experiment was performed in 3 biological replicates and graphical data was represented as Mean with SEM. $P > 0.05$ is marked as 'ns' (non-significant), $P \leq 0.05$ is marked as *, and $P \leq 0.0001$ is marked as ****.

plate reader-based quantification also showed a significantly higher fluorescence intensity for CytoRed RBC in LD-R-infected-MΦs specifically at 24 hrs pi, while MΦs incubated with dead LD-S or LD-R behaves similarly as uninfected control (Fig 5A.ii and 5A.iv). SIRPα, a specific receptor on the MΦ surface that interacts with CD47-enriched live RBCs to prevent erythrophagocytosis, was found to be significantly degraded on the surface of LD-R-infected-MΦs at 24 hrs pi, whereas a similar level of SIRPα was observed in both LD-S and LD-R infected-MΦs at 4hrs pi (Fig 5B.i.a and 5B.i.b). Western blot and flow cytometry quantification further confirmed significant degradation of SIRPα in LD-R-infected-MΦs at 24 hrs pi as compared to 4hrs pi (Fig 5B.ii.a, 5B.ii.b and 5B.iii), as opposed to CD11b, a raft-associated marker which exhibited diminished expression for both LD-S and LD-R infected-MΦs [35–37]. Notably, erythrophagocytosis has been previously reported to be a late consequence of LD-infection by post-translational downregulation of SIRPα on the surface of infected-MΦs [38].

## 7. Furin and ADAM10, acting consecutively result in the extracellular cleavage of SIRPα in LD-R-infected-MΦs

Previous observation revealed that low iron promotes augmented erythrophagocytosis in concomitance with SIRPα degradation in LD-R-infected macrophages at 24 hrs pi which is promptly possible if the extracellular domain that senses CD47-enriched live RBC is degraded. There is increasing evidence that SIRPα can be shed from the MΦ surface by the action of several proteases like MMP-9, ADAM10, and ADAM17 [38]. A recent report suggests that GI254023X, an ADAM10 inhibitor can inhibit extracellular SIRPα cleavage significantly from LD-infected-MΦs surface [38]. While the precursor form of ADAM10 (~90kDa) was observed in LD-S-infected-MΦs, an increased mature active form of ADAM10 (~68kDa) was enriched in LD-R-infected-MΦs at 24 hrs pi (Fig 6A.i and 6A.ii). Prior reports in other disease models suggest that ADAM10 gets activated by Furin [39], a proprotein serine convertase, which gets activated in iron-deprived conditions through post-translational maturation [40]. Furin undergoes post-translational maturation from the Endoplasmic Reticulum (ER), shuttles between the endosome-Golgi complex to the plasma membrane, and gets secreted as an active enzyme [41]. Although LD-S-infected-MΦs exhibited a surprisingly higher Furin expression in whole cell lysate at 24 hrs pi compared to LD-R-infected-MΦs (Fig 6B.i.a and 6B.i.b), however, immunofluorescence assays revealed that most Furin in LD-S-infected-MΦs was localized around the endosomal compartment, resembling the perinuclear localization seen in uninfected-MΦs, suggesting an inactive form (Fig 6B.ii and S3 Video). Contrarily, Furin showed diffuse cytoplasmic localization, tending toward the cell membrane, which explains

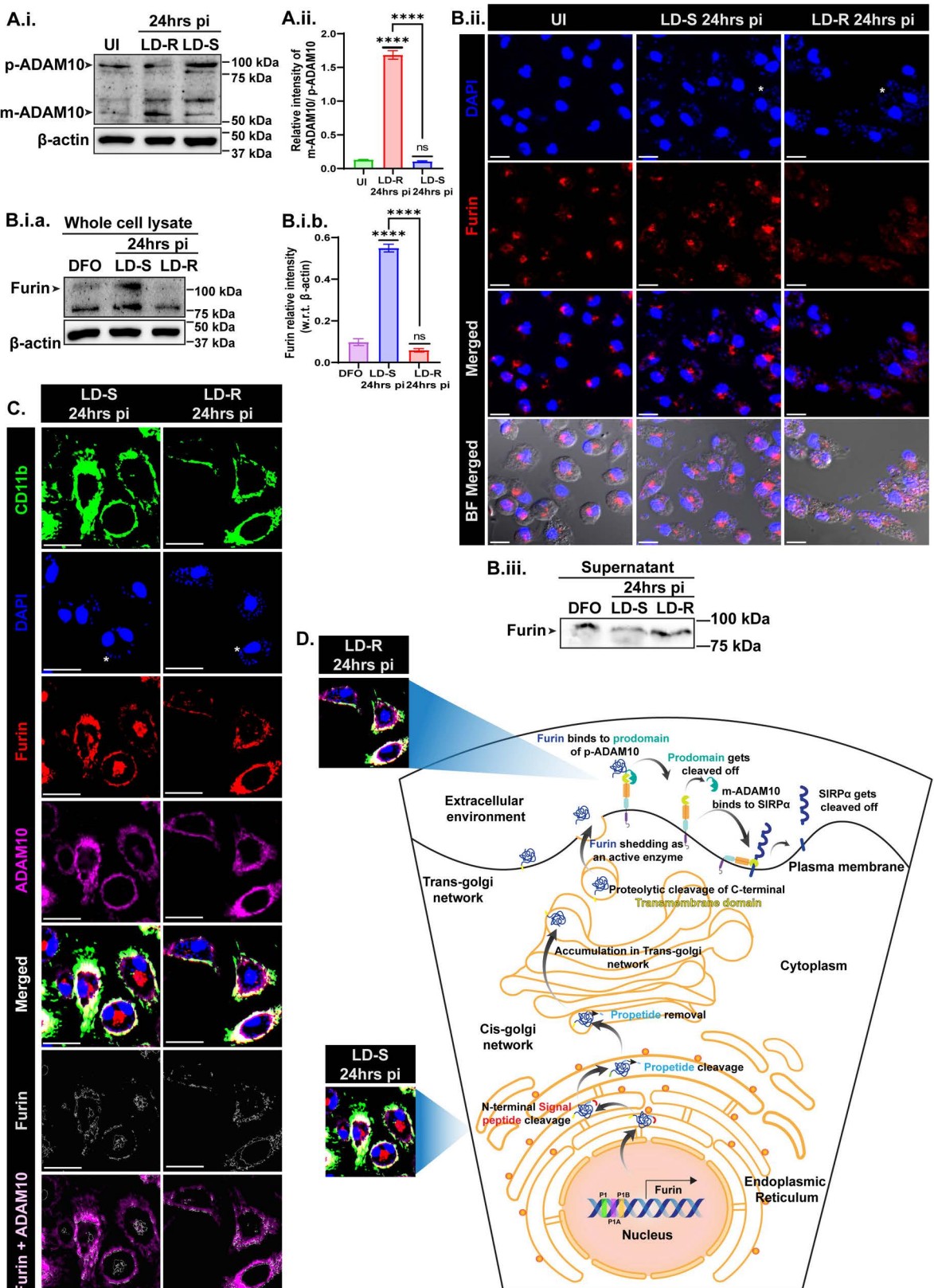

**Fig 6. Low iron activates Furin which sheds to cleave the prodomain of ADAM10 rendering it activated. (A.i.)** Western blot of whole cell lysate of uninfected-MΦs, LD-R, and LD-S infected-MΦs at 24 hrs pi showing the expression of precursor-ADAM10 (p-ADAM10,

~90 kDa) and mature-ADAM10 (m-ADAM10, ~64 kDa) keeping β-actin acts as a housekeeping control. **(A.ii.)** Bar graph representing the relative intensity of mature-ADAM10 (m-ADAM10) vs precursor-ADAM10 (p-ADAM10). Significant upregulation of relative intensity of m-ADAM10 with respect to p-ADAM10 in LD-R 24 hrs pi as compared to both UI and LD-S 24hrs pi (****, **P** ≤ 0.0001) whereas no change is observed between LD-R 24 hrs pi and UI (ns, **P** > 0.05). **(B.i.a.)** Western blot of whole cell lysate showing furin expression of DFO-treated-MΦs, LD-S, and LD-R infected-MΦs at 24 hrs pi. **(B.i.b.)** Bar graph representing the relative intensity of Furin, i.e., fold change with respect to β-actin for whole cell lysate. Significant enrichment of Furin observed in LD-S as compared to both DFO-treated control and LD-R 24 hrs pi (****, **P** ≤ 0.0001) whereas no change is observed between LD-R 24 hrs pi and DFO-treated control (ns, **P** > 0.05). Each densitometry analysis is presented as a bar graph of Mean ± SEM for 3 biological replicates. **(B.ii.)** 40X Confocal images showing the localization of furin (red) in uninfected-MΦs, LD-S, and LD-R infected-MΦs. **(B.iii.)** Western blot of furin expression in the supernatant fraction of same experiments sets as B.i. **(C.)** Confocal images showing Furin (red, 3ʳᵈ panel and edges, 6ᵗʰ panel), ADAM10 (purple), and CD11b (green) colocalization in LD-S and LD-R infected-MΦs at 24 hrs pi. One representative small nucleus of LD has been marked in (*) to show the infected-MΦs. Scale bars indicate 20μm. **(D.)** Scheme showing Furin maturation starting from Endoplasmic reticulum (ER) where it remains inactivated (observed in LD-S-infected-MΦs at 24 hrs pi) to getting shed as an active enzyme (observed in LD-R-infected-MΦs at 24 hrs pi) to cleave prodomain of p-ADAM10 rendering it activated (m-ADAM10), which in turn cleaves extracellular domain of SIRPα in LD-R-infected-MΦs at 24 hrs pi.

the low Furin level in whole cell extract of LD-R-infected-MΦs, suggesting potential Furin activation and release. Western blot analysis of supernatants from uninfected-MΦs, LD-S, and LD-R infected-MΦs revealed increased Furin shedding from LD-R-infected-MΦs at 24 hrs pi, similar to DFO-treated controls (Fig 6B.iii), likely activates ADAM10 by cleaving its prodomain. Additionally, differential permeabilization showed perinuclear and slight membrane localization of Furin in LD-S-infected-MΦs, while it was entirely membrane-localized colocalizing with ADAM10 in LD-R-infected-MΦs (Fig 6C and S3 Video).

## 8. Infection with LD-R results in severe anemia in mice by inducing enhanced erythrophagocytosis

VL is characterized by hepatosplenomegaly and anemia both of which are interlinked since an enlarged liver and spleen indicate destruction of both healthy and senescent RBC indicative of extravascular hemolytic anemia [42]. Infection with the LD-R causes increased organ-parasite load [12] with a significantly enlarged spleen as compared to infection with LD-S for both human and animal infection [3]. The ability of LD-R to induce significantly higher erythrophagocytosis in MΦs (Fig 5A.iii and 5A.iv) was further investigated in an experimental murine-model of leishmaniasis as a possible reason for severe anemia. Although, Hematoxylin-eosin (HE) stained splenic sections of uninfected, LD-S-infected-mice at 24 weeks pi revealed little or no erythrophagocytes, splenic sections of LD-R-infected-mice, exhibited a higher percentage of phagocytosed RBCs surrounded by LD-R-amastigotes (Fig 7A.i, 7A.ii and 7A.iii). Further, splenic sections incubated with DAB (which stains hemoglobin) and double stained with anti-F4/80 antibody (demarcating the red-pulp macrophages) and anti-CD47 antibody (marking the intact RBC membrane), showed both live intact RBCs (black arrow, CD47⁺/DAB⁺) and senescent RBCs (white arrow, CD47−/DAB⁺) were ingested in LD-R infected spleen. In contrast, LD-S infected splenic sections only showed engulfment of senescent RBCs (CD47−/DAB⁺), (Fig 7B). High-resolution SEM images confirmed the presence of both live intact RBCs with biconcave morphology (black arrow) and senescent RBCs with deformed morphology (red arrow) in LD-R infected splenic macrophages while only senescent RBCs (red arrow) were engulfed by LD-S infected and uninfected splenic macrophages with uninfected splenic macrophages showing the least number of engulfed RBCs (Fig 7C).

Hematology screening indicated a significant decrease in RBC, HGB, HCT, and MCV in LD-R-infected-murine-blood compared to LD-S-infected-murine-blood, with a critically low hematocrit value, suggesting severe anemia (Fig 7D.i). The other RBC indices (MCH and MCHC) do not show any significant difference between LD-S and LD-R infected-mice,

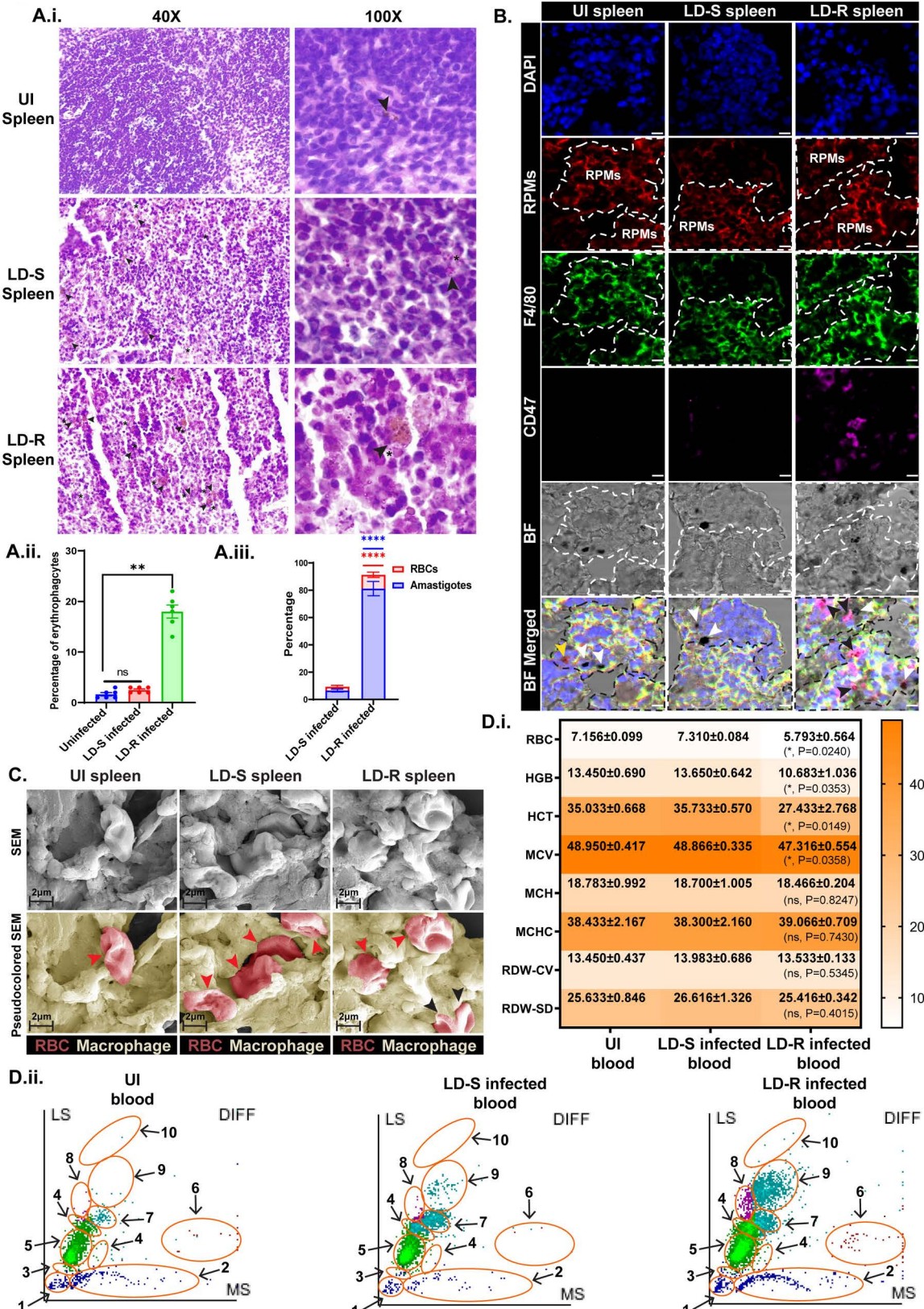

**Fig 7. BALB/c mouse infected with LD-R for 24 weeks shows traits of severe anemia. (A.i.)** Hematoxylin-eosin (HE)-stained splenic section of uninfected-BALB/c, LD-S, and LD-R-infected-BALB/c mice (from upper panel to lower panel). The left panel shows 40X images

showing a larger field of view and the right panel shows 100X images. The black arrow denotes ingested RBCs while a small nucleus of LD has been marked in (**\***) to show the infected-MΦs **(A.ii.)** Bar graph showing the total percentage of erythrophagocytes in uninfected, LD-S, and LD-R-infected splenic sections. **(A.iii.)** Stacked bar plots showing % of amastigotes and RBC engulfed by erythrophagocytes in LD-S and LD-R infected-spleen. LD-R infected spleen shows significantly higher RBCs engulfed and amastigote burden as compared to LD-S infected spleen at 24weeks pi (**\*\*\*\***, **P** ≤ 0.0001). **(B.)** Cryomicrotome sections of uninfected, LD-S, and LD-R 24weeks pi infected spleen showing autofluorescent red pulp macrophages (bright red, demarcated by a white dotted area) further stained by F4/80 (green). Live-intact RBCs were stained by CD47 antibody and ingested hemoglobin was stained by DAB. The black arrow denotes CD47$^+$/DAB$^+$ live intact RBCs, the white arrow denotes CD47$^-$/DAB$^+$ hemoglobin deposits, and the yellow arrow denotes CD47$^+$/DAB$^-$ RBC-ghosts in the merged BF image. Scale bars indicate 20 μm. **(C.)** SEM images (upper panel) of uninfected, LD-S and LD-R 24weeks pi infected murine spleen showing macrophages engulfing RBCs. The lower panel represents pseudocolored SEM where RBCs are colored in red and macrophages in yellow. The black arrow represents live RBCs with intact biconcave morphology and the red arrow represents senescent RBCs with deformed morphology. **(D.i.)** Heatmap showing the differential count of RBC-associated parameters in uninfected, LD-S, and LD-R infected-BALB/c mice. Data is represented as the Mean ± SEM where N = 6. **(D.ii.)** WBC-subpopulation scatter plots of uninfected (left panel), LD-S-infected (middle panel), and LD-R-infected (right panel) BALB/c mice-blood of one representative experimental set where the X-axis denotes the Medium angle scatter (MS) and Y-axis denotes the Low angle scatter (LS). Each subpopulation has been demarcated in (orange area) from 1–10. 1: Ghost, 2: Platelet aggregation or large RBC fragments, 3: Nucleated RBCs, 4: Abnormal lymphocytes, 5: Lymphocytes, 6: Eosinophils, 7: Left shift, 8: Monocytes, 9: Neutrophils, 10: Immature granulocytes. The dotted area signifies no significant change while the solid area denotes an area with significant changes. Each experiment was performed in 3 biological replicates and graphical data was represented as Mean with SEM. **P** > 0.05 is marked as 'ns' (non-significant), **P** ≤ 0.05 is marked as **\***, **P** ≤ 0.01 is marked as **\*\***, and **P** ≤ 0.0001 is marked as **\*\*\*\***.

indicating physiological uniformity and absence of other chronic diseases/ nutritional deficiency that could contribute to anemia. Similar RDW in both these categories also ruled out the possibility of differences in the volume and size of the RBCs contributing to anemia. WBC-differential scatter plots revealed platelet aggregation or large RBC fragments (marked as 2) in LD-R-infected-murine-blood, potentially indicating severe anemia (Fig 7D.ii). Monocytes and neutrophils also showed significant enrichment in LD-R-infected-murine-blood, suggesting inflammation persisting even at 24 weeks pi compared to LD-S-infection.

## Discussion

Our findings revolve around understanding the mechanism by which antimony-resistant *Leishmania donovani* (LD-R) has evolved as a persistent strain in due course of evolution. Mammalian hosts deploy an array of strategies to eliminate LD, while LD has evolved sophisticated yet diverse mechanisms to counteract the host-defense strategies during intracellular proliferation. Our observation pointed out that high organ-parasite load linked with LD-R-infection [3] does not rely only on initial infectivity as commonly believed to date [3,13], rather indicative of a selective survival advantage for LD-R (Fig 1). To understand how LD-R might have evolved as a persister strain, the strategy for evading the anti-leishmanial action of antimony has to be decoded first. The anti-leishmanial action of pentavalent antimonials and the initial host-defense strategy include triggering of ROS which causes DNA damage [43,44]. Logical interpretation leading to enhanced survival of LD-R lines inside the host is their capability to prevent host-derived oxidative outbursts. Contrarily, LD-R instead of suppressing/ delaying ROS production, was found to elicit ROS for its survival benefits.

LD-R, as opposed to LD-S, showed an inherent metabolic shift from Glycolysis towards the pentose phosphate pathway resulting in enriched reducing intermediates which allows them to withstand higher oxidative stress (Fig 2). A previous report suggests that drug-induced oxidative stress shifts central carbon metabolism to the PPP in intracellular-LD [45]. The fact that this induction of initial high ROS is observed only for LD-isolates showing primary unresponsiveness only to antimony but not towards other drugs like Amp-B, probably indicates this might be a common occurrence restricted to LD-R field isolates. It's worth noting that the delayed generation of ROS in response to LD-S-infection has been linked to surface lipophosphoglycan [8]. Considering that LD-R expresses unique surface glycans compared

to LD-S [46], this might contribute to the varied induction of ROS in response to LD-S or LD-R-infection.

This posed an inherent question how is this initial oxidative outburst advantageous in providing survival benefits to LD-R? Notably, a previous study on *Trypanosoma cruzi* has shown elevated ROS promotes parasite proliferation inside host MΦs [47,48]. Our findings suggest that initial ROS elevation prompts Heme-oxygenase-1 (HO-1) upregulation (Fig 3), catalyzing heme degradation into bioavailable $Fe^{2+}$, which is crucial for heme-auxotrophic-LD parasites lacking iron-synthesizing machinery [62]. Notably, p50/c-Rel-dependent activation of HO-1 was not observed for LD-S-infected-MΦs (Fig 3). Prior reports have suggested that while LD-S-infection leads to degradation of host transcription factors, including p50/c-Rel [49,50], LD-R's interaction with host-MΦs results in TLR-dependent activation of p50/c-Rel singling [23,46].

Iron is essential for DNA replication, ATP synthesis, and mitochondrial respiration [63–65] and thus elevated iron and its subsequent mobilization within LD-R-PV will fulfill their iron requirements for rapid proliferation. To date, most studies most studies illustrated how LD prevents iron-efflux as a strategy to withhold iron for themselves [31] with little or no knowledge of understanding how iron or heme is trafficked inside the PV, despite iron/heme transporters being well-documented in the *Leishmania* genome [51–53]. It is plausible that the LD-R genome contrary to its sensitive counterparts may harbor specific amplification of iron transporter and metabolizing proteins such as LDBPK_313190 and LDBPK_366760 [54], although further work is necessary to validate this hypothesis.

Our results confirmed an increased influx of $Fe^{2+}$ into LD-R-PV, accompanied by the upregulation and targeted redistribution of the MΦ membrane iron-exporter, Ferroportin, around LD-R-PV during phagosomal compartment formation, facilitating the accumulation of excess bio-available iron for LD-R-amastigotes (Fig 4 **and** S2 Video). A previous report on *Salmonella typhimurium* has suggested a similar inward reorientation of Ferroportin responsible for providing iron in *Salmonella*-containing vacuole rather than its conventional role in iron-efflux from infected hosts [32]. Interestingly, Hepcidin, a peptide hormone produced in the liver, has been reported to degrade Ferroportin only when exposed to MΦ surface [32]. Reduced $Fe^{2+}$ accumulation in LD-R-PV in Hepcidin pre-treated-MΦs due to minimal Ferroportin accumulation around LD-R-PV as opposed to Hepcidin post-treatment convincingly proved that LD-R has devised a strategy to reorient Ferroportin around LD-R-PV to drive excess iron inside its PV (Figs 4 **and** S3).

Increased influx of iron inside LD-R-PV results in a drop in cytoplasmic $Fe^{2+}$ which in turn promotes non-canonical activation of NRF2 in the LD-R-infected-MΦs leading to anti-oxidative response to bring down inflated ROS at a later time as LD-R-infection progresses (Figs 2 **and** S4).

Once most of the iron is utilized by LD-R, a continued iron supply will still be required to maintain its ongoing proliferation. MΦs are the key hub for recycling senescent RBCs to produce iron through erythrophagocytosis. Previous reports have shown significant erythrophagocytosis in LD-infected-MΦs and mice [28]. These observations and the clinical report of severe anemia in LD-R-infected-patients [55,56], propelled us to investigate the status of erythrophagoctosis as a possible cause of anemia in LD-infection. Cross-talk between CD47, enriched in live RBCs, and SIRPα, expressed in the MΦ membrane is the critical determinant of erythrophagocytosis [28,57]. When CD47-enriched live RBCs come in contact with intact SIRPα, it elicits a 'don't eat me' signal thus the macrophage discriminately phagocytoses senescent RBCs leaving behind the live ones. Our results revealed SIRPα degradation from the surface of LD-R-infected-MΦs leading to loss of discriminatory surveillance thus promoting augmented erythrophagocytosis (Fig 5). Our study also deciphered a possible connection

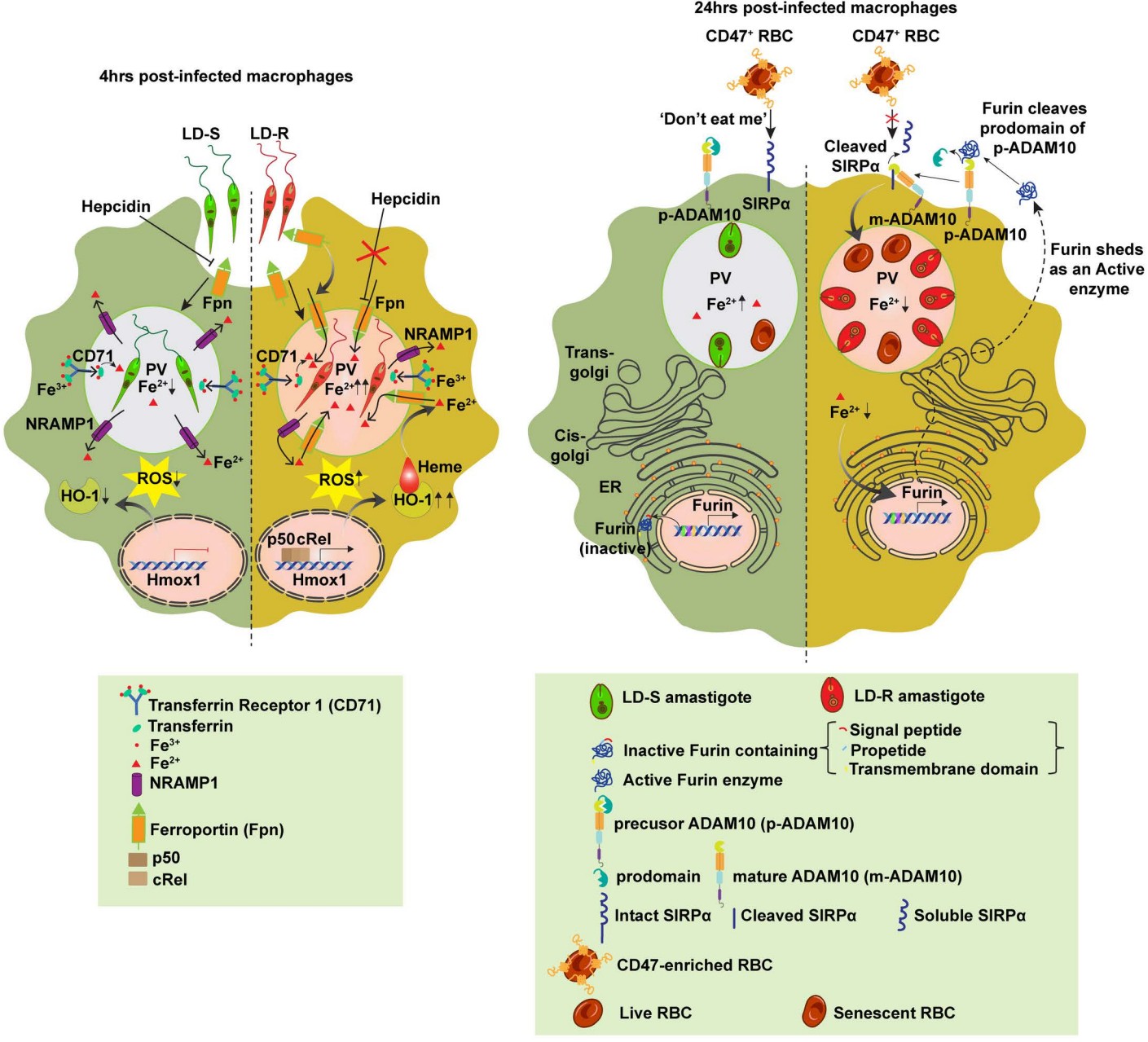

**Fig 8. Representative scheme of the downstream signaling upon LD-S vs LD-R infection at 4 hrs and 24 hrs pi.** At 4 hrs pi (left panel), LD-R triggers ROS production which promotes nuclear translocation of p50 and c-Rel. p50/c-Rel binds to the promoter of Heme-oxygenase 1 and activates it. HO-1 protein catalyzes the breakdown of heme into iron ($Fe^{2+}$). LD-R reshuffles Ferroportin (Fpn) around LD-R-PV while Fpn remains in MΦ surface in LD-S infection. NRAMP1 is down-regulated in LD-R-PV as compared to LD-S-PV while there is a uniform expression of Transferrin receptor 1 (CD71) in both cases. All these factors drive excess iron produced from HO-1 inside LD-R-PV as compared to LD-S-PV. At 24 hrs pi (right panel), LD-R proliferates at a rapid rate utilizing this excess iron inside LD-R-PV resulting in a drop of the iron level. This iron depletion condition promotes Furin shedding as an active enzyme which then binds to the prodomain of p-ADAM10 present in the macrophage membrane and cleaves it, rendering it activated (m-ADAM10). m-ADAM10 binds to the extracellular domain of SIRPα and cleaves it, breaking the discriminatory signal of CD47-enriched live RBCs or CD47-less senescent RBCs, resulting in aggravated erythrophagocytosis of both live and senescent RBCs. Contrarily, in LD-S infection, Furin remains mostly around perinuclear space which is the inactive form and cannot interact with membrane-associated p-ADAM10. Thus, SIRPα remains intact and elicits a 'Don't eat me' signal upon interacting with CD47-enriched live RBCs resulting in controlled erythrophagocytosis.

between low cytoplasmic iron in LD-R-infected-MΦs at 24 hrs pi with subsequent loss of SIRPα. Iron deficiency has been previously reported to activate Furin [40], a proconvertase that reaches out to the macrophage membrane and cleaves the prodomain of ADAM10 to activate it [41]. Mature-ADAM10 has been recently reported to promote extracellular proteolysis of SIRPα [58]. Connecting these dots, we were able to show that while in LD-S-infected-macrophages, Furin was mostly restricted to a perinuclear localization, in response to LD-R-infection, they were mostly present on the MΦ surface co-localizing with ADAM10 to activate it (Fig 6 **and** S3 Video). Finally, our results comparing LD-S and LD-R infection in a murine-model of VL-induced anemia validated a higher percentage of erythrophagocytosis in LD-R-infected-mice, with a higher frequency of ingested RBCs by LD-R-amastigotes (Fig 7). Blood parameters- RBC, HGB, and HCT further confirmed traits of severe anemia in LD-R-infected-mice. This study deciphers downstream pathways starting from initial infection to aggressive pathogenesis leading to chronic hepatosplenomegaly and severe anemia in response to LD-R-infection which was hitherto unexplored as summed up in Fig 8. Our findings shed light on refining therapeutic strategies to address clinical antimony-resistant LD-infection. It also explains the rapid emergence of unresponsiveness in LD and related heme-auxotrophic pathogens to drugs reliant on oxidative outbursts.

## Materials and methods

### I. Ethics statement

The use of mice was approved by the Institutional Animal Ethics Committees of the Indian Institute of Technology, Kharagpur, India. All animal experiments were performed according to the National Regulatory Guidelines issued by biosafety clearance number IE-01/BM-SMST/1.23. a.

### II. Reagents and antibodies

Lipofectamine 2000 Reagent (11668019) was purchased from Invitrogen. Deferoxamine mesylate (DFO), an iron chelator, was purchased from Sigma (D9533). NADPH (481973) and NADP (481972) were purchased from Merck. Giemsa stain solution (72402) was purchased from SRL. DAPI (TC229) was purchased from Himedia. Fluoromount-G (00-4958-02) was obtained from Invitrogen. Harris hematoxylin and eosin were obtained from Merck. SSG is a kind gift from Albert David, Kolkata. Lysotracker Red (L7528) and Calcein-AM (C3099) were purchased from Invitrogen. Antibodies against Rab5a (E6N8S), NRF2 (D1Z9C), Histone (H3) (D1H2), p50 (D4P4D), c-Rel (D4Y6M), FTH-1 (D1D4), CD11b (E6E1M), F4/80 (71299), GAPDH (D16H11) and β-actin were purchased from Cell Signaling Technologies (CST). Anti-Mouse CD11b-Alexa Fluor 488 is purchased from Invitrogen (53-0112-82). Antibodies against p47phox (sc-17844) and p50 (sc-8414) were purchased from Santa Cruz Biotechnology. Antibodies against p62/SQSTM1 (MA5-27800) and Hmox1 (PA5-88074) were purchased from Invitrogen. Antibodies against CD71/Transferrin Receptor (A5865) and Rab5a (A1180) were purchased from Abclonal. Antibody against NRAMP1 was purchased from Santa Cruz Biotechnology (sc-398036). Antibody against Furin was purchased from Abclonal (A5043) and CST (E1W40). Antibody against ADAM10 was purchased from Abclonal (A10438) and Invitrogen (MA5-23867). Anti-Mouse CD172a (SIRP alpha)-APC was purchased from Invitrogen (17-1721-82). Ferroportin/ SLC40A1 antibody was purchased from Novus Biologicals (NBP2-45356). Antibody against CD47 was purchased from Invitrogen (PA5-114984). HRP-conjugated anti-mouse secondary antibody (57404) was purchased from SRL and anti-rabbit secondary antibodies were purchased from Sigma (A8275). Alexa Fluor 488 goat anti-mouse (A11001), Alexa-fluor 594 goat anti-mouse (A11005), Alexa-fluor 488 goat

anti-rabbit (A11008), and Alexa-fluor 594 goat anti-rabbit (A11037) secondary antibodies were purchased from Invitrogen. Anti-rat IgG (H+L) Alexa Fluor 647 antibody (4418S) was purchased from CST.

### III. Mice maintenance

BALB/c mice (*Mus musculus*) were maintained and bred under pathogen-free conditions.

### IV. Macrophage (MΦ) isolation

Mouse peritoneal MΦs were harvested from BALB/c mice by lavage, 48hours (hrs) after intra-peritoneal injection of 3% soluble starch (SRL). The adherent peritoneal exudate cells (PECs) were defined as MΦs for convenience. MΦs were plated on sterile 12 mm round coverslips (Bluestar) in 24-well plate or 35mm dish (Tarsons) or 6-well plate at a density of $1\times10^5$ cells/ coverslip or $1\times10^6$ cells respectively in RPMI 1640 medium (GIBCO) supplemented with 10% FBS (GIBCO) and 100 U PenStrep (GIBCO) (RPMI complete medium) as and when required. The cells were kept for 48hrs at 37ºC and 5% $CO_2$ to adhere before infection.

### V. Parasites maintenance

*Leishmania donovani* clinical isolates used in this study were as follows: MHOM/IN/83/AG83, MHOM/IN/09/BHU777/0, MHOM/IN/10/BHU814/1, MHOM/IN/09/BHU575/0 (kind gift from Dr. Syamal Roy, CSIR-IICB, India), and MHOM/NP/03/D10. DD8 is primarily an antimony-sensitive strain while the Amphotericin-B resistant DD8 strain is laboratory-generated and is a kind gift from Dr. Arun Kumar Halder, CDRI, Lucknow, India. D10 [59] is obtained from the Laboratory of Molecular Immunology, Graduate School of Agricultural and Life Sciences, The University of Tokyo, Bunkyo-ku, Tokyo, Japan. Transgenic LD-S-GFP, LD-R-GFP, and LD-R-RFP lines were generated in the lab. All the isolates were maintained in BALB/c mice. Promastigotes were maintained in a 22ºC auto shaker incubator in M199 medium supplemented with 10% FBS. As per experimental requirements, promastigotes were treated with 100μM $H_2O_2$ followed by 6-Aminonicotinamide for generating NADPH-exhausted LD [18].

### VI. *In-vivo* and *in-vitro* infection

For *in-vivo* infection, metacyclic promastigotes ($1\times10^7$) were pelleted and dissolved in 200 μl sterile PBS, and intravenous infection was performed in the tail vein of 4–6 weeks old female BALB/c mice using an insulin syringe and the mice were sacrificed at 28days pi (for splenic amastigote load) or 24weeks pi (for establishing anemia model) as per experimental requirements. Infected mice (N=3) (infected with LD-S-GFP (Group:1), LD-R (Group:2), LD-S-GFP: LD-R in 50:50 ratio (Group:3), LD-S-GFP: LD-R in 80:20 ratio (Group:4), LD-S-GFP: LD-S in 50:50 ratio (Group:5)) were sacrificed after 28 days pi and the spleen was isolated, weighed, and cut into 3 sections. The first section weighing ~100mg for each group was used for quantifying *amastin* expression by qRT-PCR to enumerate splenic amastigote load [60]. The second section was further divided into two parts: one portion was stamp-smeared and stained with Giemsa, and observed in the bright-field microscope to determine the organ parasite load expressed as LDU (LDU= amastigote per 1,000 nucleated cells × organ weight in gram), another portion was fixed, cryosectioned, stained with DAPI and observed under Fluorescence microscope. The third section was macerated and kept in M199 medium with 10% FCS to allow differentiation of LD-amastigotes to promastigotes and the GFP-positive promastigote population from 1st passage was enumerated by flow cytometry. To establish the anemia model, infected mice were sacrificed at 24 weeks pi keeping uninfected mice as control (N = 3). Before sacrificing, blood was drawn from the cardiac puncture under anesthesia with

isoflurane, and collected in an EDTA-vacutainer. These samples were run in Erba H560 Fully Automated 5-part Hematology Analyser and the hematological parameters were noted. The mice were then sacrificed by cervical dislocation and the spleen was separated for further analysis of the splenic parasite burden and erythrophagocytosis monitoring.

For *in-vitro* infection, peritoneal MΦs from BALB/c mice were infected with LD strains at a ratio of 1:10 (macrophage: parasite) for 4hrs (considered as 0hrs) infection followed by washing, thereafter incubated for a specific time according to experimental requirements.

### VII. Parasitophorous vacuole (PV) isolation

PV was isolated following the method described previously [61]. Briefly, infected cells were gently scraped off and resuspended in homogenization buffer (20mM HEPES, 0.5mM EGTA, 0.25M sucrose, and 0.1% gelatin supplemented with protease inhibitor cocktail). Lysis was performed by repeated passage (5–20 times) through two 1ml syringes with 23G needles. The mixture was then centrifuged at $200 \times g$ for 10minutes (mins) to pellet intact cells and nucleus. The low-speed supernatant was then loaded onto a discontinuous sucrose gradient consisting of 3ml 60% sucrose, 3 ml 40% sucrose, and 3 ml 20% sucrose in HEPES-buffered saline (30 mM HEPES, 100 mM NaCl, 0.5 mM CaCl, 0.5 mM $MgCl_2$, pH 7) in 15ml falcon tube and centrifuged at $700 \times g$ for 25mins at 4ºC. The parasitophorous vacuoles (PV) were harvested from the 40–60% interface carefully. It was then centrifuged and pelleted at $12,000 \times g$ for 25mins. The supernatant was discarded and the pellet fraction is PV.

### VIII. Measurement of intracellular ROS production

Peritoneal MΦs were plated in a 96-well plate at a density of 20000 cells/well and allowed to adhere for 48hrs. MΦs were infected with metacyclic promastigotes at a ratio of 1:10 (macrophage: parasite) for 1–8 hrs. MΦs were loaded with $H_2DCFDA$ (2',7'-Dichlorodihydrofluorescein diacetate) at 10 μM final concentration and kept for 45mins at 37ºC 5% $CO_2$ followed by washing in HBSS. DCF fluorescence was considered as a read-out of intracellular ROS. Area-scan readings were taken using a Biotek Cytation 5 imaging reader at Ex/Em:492–495/517–527nm. Each experimental sets were performed in triplicates. Similar experimental sets were performed for live-cell flow cytometry estimation of DCF-FITC-A positive population. Data were analyzed in FlowJo v10.

### IX. Quantification of NADPH

NADPH is first extracted from pelleted LD-R and LD-S promastigotes using a method as described elsewhere [62]. Briefly Solvent A (40:40:20 acetonitrile: methanol: water supplemented with 0.1M formic acid) was added to the parasite pellet, vortexed for 10seconds (secs), and allowed to rest on ice for 3mins. For each 100 μl of solvent A, 8.7 μl of solvent B (15% $NH_4HCO_3$ in water (w:v) precooled on ice) was then added and vortexed to neutralize the sample. The mixture was kept in dry ice for 20mins. Samples were then centrifuged at $16,000 \times g$ for 15 mins at 4ºC and supernatant was taken for LC-MS analysis.

### X. Metabolomics analysis

Mid-log phase promastigotes were considered for metabolite extraction following the protocol described in t'Kindt *et al.* [63]. The Waters 2695 Separation Module was used coupled with Quattro micro API analyzer (quadrupole mass analyzer) machine with a C18 column as the stationary phase and a methanol-water mixture with increasing concentration over time as the mobile phase. The MS1 spectral data were generated in dual ion mode using Electron Spray ionization. Generated Water Raw files were converted to mzXML files, an open mass

spectrometry data format, using the msconvert tool from ProteoWizard [64] filtering only positive ions. A GUI-based Mass Spectrometry Data Processing Engine called El-Maven v0.12.0 [65] was used for subsequent analysis. Peak alignment was performed using the OBI-Warp alignment algorithm and Savitzky–Golay filter for ESI smoothening. The peak annotation was done using the estimated m/z of the selected compounds by their compound formula with a 100-ppm error and 5 best peaks for each group were selected. From the result annotation, the sum of the intensities was used for each metabolite and any peak with a high Retention Time (RT) difference was manually excluded from the analysis.

## XI. Gene-Set Enrichment Analysis (GSEA)

GSEA was performed leveraging publicly available Microarray data from NCBI GEO data set: GSE144659 to differentially compare clinical LD-R-isolates (BHU575, BHU814) and LD-S-isolates (AG83, BHU777). A Group-Wise Mean Normalisation is performed to make the data comparable for all the samples. A Multi-Dimensional Scaling is performed to generate a PCA plot. Using the generated normalized data, and merging it with Glycolysis and Pentose phosphate pathway gene sets obtained from KEGG, the GSEA analysis is performed using the GSEA GUI software [66,67].

## XII. RNAseq analysis

Differential Expression Analysis (DGE) was performed for LD-S 4hrs, LD-S 24 hrs, LD-R 4hrs, and LD-R 24 hrs post-infected (pi) MΦs keeping uninfected (UI) MΦs as control, using the RNA-Sequencing read count data with DESeq2 library on R [68]. RNA-Sequencing was performed utilizing external service by Bionivid Project (GSE279792, https://www.ncbi.nlm.nih.gov/geo/query/acc.cgi?acc=GSE279792). The results were represented as a volcano plot by using a cutoff of $\log_2$(Fold-Change) $\geq 0.6$ for upregulated genes and $\log_2$(Fold-Change) $\leq -0.6$ for downregulated genes [69,70], and with Benjamini-Hochberg adjusted p-value < 0.05. Normalized counts (N = 2) were obtained from the DESeq2 package and averaged normalized counts were used to develop heatmaps of a few selected genes using gplots and RColorBrewer. To decode the differential polarity of MΦs infected with LD-S vs LD-R at 4hrs and 24 hrs pi, the gene sets contributing towards M1 and M2 phenotypes were obtained from the previous report [71] and tallied with our RNAseq data. Score of M1/ M2 gene cluster = Sum of respective M1/M2 $\log_2$(fold change) of genes with adjusted *p-value* < 0.05.

The difference in these scores indicates the inclination of the polarity of MΦs toward M1 or M2 polarization (−ve indicating M2 polarization and +ve indicating M1 polarization).

$$Score\ of\ M1/\ M2\ polarization\ gene\ set$$
$$= Sum\ of\ resepective\ Log2FoldChange\ of\ genes\ with\ adj\ p-value < 0.05$$

$$Final\ determinant\ score = Score\ from\ M1\ gene\ Set - Score\ from\ M2\ gene\ Set$$

## XIII. Calcein-AM-based estimation of Labile iron pool (LIP)

Labile iron pool status was estimated using Calcein-AM as described previously [24,25]. Calcein in its unbound form fluoresces, while in the presence of iron, Calcein fluorescence gets quenched. Briefly, $2 \times 10^5$ peritoneal MΦs were plated in Confocal dishes for 48hrs followed by infection with LD promastigotes at a specific time point. MΦs were then loaded with 0.5μM working concentration of Calcein-AM (Ex/Em:494/517nm) for 30 mins at 37ºC. It was washed properly with 1XPBS after which MΦs were exposed to 50 nM Lysotracker Red (Ex/

Em:577/590nm) to demarcate the parasitophorous vacuole for 1hr at 37°C followed by washing in 1XPBS. MΦs were then incubated with DAPI in 1XPBS followed by washing and taken for Confocal microscopy for live cell imaging.

## XIV. Quantification of cytoplasmic and intraphagosomal iron

Iron concentration was quantified by colorimetric ferrozine-based assays as described previously [72]. For whole intracellular quantification of iron, infected cells were scraped off. Phagosomes of infected cells were isolated for intra-phagosomal iron quantification and cytoplasmic fractions were isolated for cytoplasmic iron quantification as mentioned earlier. Briefly, cytoplasm/ phagosomes were lysed by application of 200μl 50mM NaOH + 200μl 10mM HCl + 200μl iron releasing reagent (a freshly mixed solution of equal volumes of 1.4M HCl and 4.5% (w/v) $KMnO_4$ in $H_2O$). The mixtures were kept in the dark and incubated for 2hr at 60°C within a fume hood. 60 μl of iron detection reagent (6.5mM ferrozine, 6.5mM neocuproine, 2.5M ammonium acetate, and 1M ascorbic acid dissolved in water) was added after the samples were cooled down and incubated for another 30 mins at room temperature. 280 μl of solution was added to a 96-well plate and absorbance was measured at 550 nm. Iron concentrations were determined using known $FeCl_3$ standards. Each experimental sets were performed in triplicates.

## XV. Plasmids, transfection, and luciferase reporter assay

Using the Eukaryotic Promoter Database (EPD) heme oxygenase-1 promoters were determined. Using the PROMO tool, the transcription factor binding sites were determined for p50 and c-Rel. Murine HO-1 promoters (−1385/+137), 1522 bp using primers 5'-AAGGTACCTGAGGCTGGAGAGATGGCC-3' and 3'- TAAAAGCTTCACCGGACTG GGCTAGTTCAG-5' were PCR amplified and cloned in promoterless PGL3 enhancer empty vector (Promega, E1771) at the upstream of luciferase gene. The whole HO-1 promoter was separated into two halves: Site A$^{-/-}$ (−4/−635) and Site B$^{-/-}$ (−636/−1377). Using NEBase Changer tool, two truncated promoter primers (Site A$^{-/-}$ and Site B$^{-/-}$) were designed: Site A-/- (5'-TCAGATTCCCCACCTGTA-3' and 3'-GGTACCTTTATCGATAGAGAAATG-5') and Site B-/- (5'- GCTCACGGTCTCCAGTCG-3' and 3'- GCTGGAGGTTGAAGTGTTC-5'). Two sites were mutated following the Q5 Site-Directed Mutagenesis protocol. For difficulty in achieving high efficiency transfection in primary murine macrophages, RAW 264.7 macrophage cell lines were transiently transfected with HO-1+PGL3, SiteA$^{-/-}$HO-1+PGL3, and SiteB$^{-/-}$HO-1+PGL3 plasmids using Lipofectamine 2000 reagent (Invitrogen) for 6hrs, washed and incubated for another 12hrs followed by infection with LD-S or LD-R or kept uninfected. Luciferase activity was measured in cell extracts using a Dual-Luciferase Reporter Assay Kit (Promega).

## XVI. Western blot analysis

Following infection and other treatments, peritoneal MΦs were scraped off in ice-cold PBS followed by centrifugation and lysed in 2X SDS sample loading buffer consisting of 20% glycerol, 10% β-mercaptoethanol, 4% SDS, 0.125M Tris HCl, 0.004% Bromophenol blue, pH = 6.8. Samples were boiled at 95°C for 10mins and subjected to SDS-PAGE. Resolved proteins were transferred in a nitrocellulose membrane (Bio-Rad) using a Semi-dry transfer apparatus (Invitrogen). Membranes were blocked in blocking solution (5% milk in 0.05%Tween20/ PBS) for 30mins at room temperature followed by probing in primary antibodies diluted in blocking solution overnight at 4°C in an orbital shaker. Membranes were then incubated with the following HRP conjugated goat anti-rabbit IgG or goat anti-mouse IgG secondary

antibody. The substrate working solution was using SuperSignal West Pico PLUS Chemi-luminescent Substrate (Thermo Scientific). The membranes were incubated with the substrate working solution for developing and the chemiluminescence blot images were taken in Chemidoc MP Imaging System (Bio-Rad). GAPDH or β-actin and Histone (H3) were used as a positive control for cytoplasmic fraction or whole cell lysate and nuclear fraction respectively.

## XVII.  Immunofluorescence

For immunofluorescence studies, infected, uninfected, or treated MΦs plated in glass-coverslips were fixed in 2% paraformaldehyde for 5–10 mins. The fixative was aspirated and neutralized with 0.1M glycine/PBS for 5mins followed by permeabilization in 0.2% Triton in 1XPBS for 20 mins on an orbital shaker. Subsequently, MΦs were blocked with blocking solution (2% BSA in 0.2% Triton in 1XPBS) for 20 mins on an orbital shaker at 4ºC. MΦs were then incubated in primary antibodies diluted in blocking solution against specific proteins for 1hr at room temperature on an orbital shaker. MΦs were then washed thoroughly for 3 × 5mins with 0.1% Triton/PBS. For differential permeabilization assays, permeabilization-free (without detergent) labeling was performed for membrane protein (in this case CD11b and ADAM10) followed by permeabilization (0.1% Triton in 1XPBS) and labeling with antibodies against the intracellular protein (in this case Furin). Finally, Alexa-Fluor labeled secondary antibodies diluted in blocking solution were loaded onto the MΦs and incubated for 1hr at room temperature on an orbital shaker followed by washing 3 × 5mins with 0.1% Triton/PBS. The coverslips were then mounted using Fluoromount+DAPI in glass slides and viewed in a Confocal microscope using an oil immersion 63X objective.

## XVIII.  *In-vitro* monitoring of erythrophagocytosis

Erythrophagocytosis was monitored using a method used elsewhere [28]. Briefly, GFP-tagged LD promastigotes were used for infection on peritoneal MΦs (1:10 macrophage: parasite) seeded on 16-well chamber slide glass for specific time points (4 hrs or 24 hrs). MΦs were then incubated with freshly prepared 100 μl Cyto-Red RBC and kept for 4hrs at 37ºC, 5% $CO_2$ followed by being washed and monitored by live cell imaging.

## XIX.  RNA isolation from spleen and amastin quantification by qRT-PCR

~100 mg of splenic sample were homogenized using a micro-pestle and resuspended in RNAiso Plus (Takara). RNA was isolated using the manufacturer's protocol and the quality and quantity were estimated in a NanoDrop Lite spectrophotometer (Thermo Scientific). Using ProtoScript First Strand cDNA Synthesis Kit (New England BioLabs), RNA is denatured and cDNA was synthesized. From the diluted cDNA product, using PowerUp SYBR Green Master Mix and Forward primer: 5'-GTGCATCGTGTTCATGTTCC-3'; and Reverse primer: 3'-GGGCGGTAGTCGTAATTGTT-5' were subjected to qRT-PCR for amplifying Amastin in QuantStudio 5 (Applied Biosystems) in triplicates. Finally, the amplification plots were generated and analyzed using QuantStudio Design & Analysis Software v1.5.2 (Applied Biosystems). The ΔCt and ΔΔCt for Amastin were calculated with respect to Murine β-actin (Forward primer: 5'-AGAGGGAAATCGTGCGTGAC-3', and Reverse primer: 3'-CAATAGTGATGACCTGGCCGT-5').

## XX.  Flow cytometry

Metacyclic promastigotes were sorted using Beckman Coulter Cytoflex srt by gating *FSC-A* X *SSC-A* where FSC^low represents the metacyclic population and FSC^high represents the procyclic

population [13]. Next, a portion of these sorted metacyclic promastigotes was further checked on the Flow cytometer capturing 20,000 events which is represented in Figs 1A.i, 1A.ii and S1A signifying uniform metacyclics between LD-S and LD-R for performing downstream experimental studies. While GFP-positive P1 populations were enumerated in the Flow cytometer (BD LSRFortessa Cell Analyzer). Briefly, 5ml P1 promastigote culture differentiated from spleen-macerates was resuspended in 1ml FACS buffer and run in FACS and the data is recorded in logarithmic scale. For the SIRPα+-population enumeration in infected MΦs, $1\times10^6$ cells were stained by labeling with Anti-Mouse CD172a (SIRP alpha)-APC and Anti-Mouse CD11b-Alexa Fluor 488 followed by washing and fixing with 2% PFA. The cells were resuspended in 500 μl ice-cold FACS buffer and run in the Flow cytometer. All analyses were performed in FlowJo v10.

## XXI. Histology staining

Spleen collected at the time of sacrificing the BALB/c mice were fixed with 20% buffered formalin overnight and a representative section was subjected to tissue processing, and embedded in paraffin the next day. Once the paraffin mold was completely dried, the tissue block was cut at 4 μm thickness in Microm HM 315 Microtome and placed in Mayer's egg albumin-positive slides. Once fully dried, the slides were deparaffinized using xylene followed by dehydration, and then stained with hematoxylin for 30 secs. It was then rinsed with running tap water for 30 mins followed by eosin staining for 10secs. It was rinsed again under running tap water for 20 mins followed by dehydration and mounting in D.P.X. (Sigma) and observed under the microscope.

For cryosectioning, the spleens were washed briefly with PBS followed by fixation using 2% PFA for 6hrs. The spleen samples were cryopreserved using a sucrose gradient starting from 15% sucrose in PBS until tissue sinks (6–12 hrs) further followed by 30% sucrose in PBS until tissue sinks (6–12 hrs). The spleen samples were then embedded in M-FREEZE Cryoembedding media in cryomold overnight. Using a cryo-microtome (Leica CM 1520), the spleen samples were cut into 4μm thin sections and collected onto polylysine-coated coverslips. These sections were stored at −20°C until further staining.

## XXII. DAB and immunofluorescent staining

3,3'-Diaminobenzidine tetrahydrochloride hydrate (DAB) is used to stain RBCs containing hemoglobin. Spleen sections were washed using PBS and incubated in DAB solution (0.6 mg/ml in Tris-buffered saline) containing Ni(II)SO$_4$ (10 mM) for 10 minutes, followed by adding 1 μl 30% hydrogen peroxide and incubating for another 10 minutes to develop color. The sections were then washed with PBS to prevent further color development. DAB-stained sections were further processed for immunofluorescent staining using CD47 and F4/80; starting from permeabilization using 0.02% Triton followed by all downstream steps of immunofluorescent staining mentioned before.

## XXIII. Scanning electron microscopy (SEM)

The spleen sections adhered in the polylysine-coated coverslips were further processed for SEM. Briefly, the spleen samples were washed with PBS followed by increasing ethanol gradient dehydration using 70%, 80%, 90%, 95%, and finally 100% twice each for 10 mins. After final incubation, ethanol is removed and completely airdried. Dry coverslips were stored in a vacuum desiccator till imaging. On the imaging day, sections were sputtered by gold using Quorum Q150R ES and proceeded for imaging using Field Emission Zeiss Sigma 300 VP scanning electron microscopy unit.

## Supporting information

**S1 Table.** **(A)** Table showing the list of Glycolytic genes presented in the heatmap of Fig 2E.i with their respective enrichment scores. **(B)** Table showing the list of Pentose phosphate pathway (PPP) genes presented in the heatmap of Fig 2E.ii with their respective enrichment scores. (DOCX)

**S1 Fig. Antimony-resistant phenotype coincides with higher parasite burden if metacyclogenesis is kept as a constant factor (A.)** Representative flow cytometer images (BD LSRFortessaTM Cell Analyzer) showing the % of metacyclic promastigote in AG83, BHU777 (7th-day culture); BHU814, BHU575 (5th-day culture) after uniform sorting in Beckman Coulter Cytoflex srt. FSC$^{low}$ (left) represents the metacyclic population and FSC$^{high}$ (right) represents the procyclic population. **(B.i.)** Giemsa-stained images of an equal number of AG83, BHU777, BHU814, and BHU575 metacyclic promastigote infected-MΦs at 4 hrs, 24 hrs, 48 hrs, and 72 hrs pi. One representative small nucleus of LD has been marked in (**\***) to show the infected-MΦs. **(B.ii.)** No. of infected MΦs/100 MΦs at 4 hrs pi were enumerated. Each data is the mean of three individual sets. **(B.iii.)** Amastigotes/100 MΦs at 4 hrs, 24 hrs, 48 hrs, and 72 hrs pi were calculated taking different fields, and the data represents the SEM of three independent experiments. **(C.i.)** Confocal images of macerated spleen sample of LD-S-GFP: LD-S showing the GFP-amastigote load. Scale bars indicate 20 μm. **(C.ii.)** % of the GFP-positive population was enumerated from FACS to denote the load of LD-S-GFP. The right-most panel shows the ancestry of each analysis. **(D.)** The dose-response curve of clonal LD lines derived from LD-S-GFP, LD-R, LD-S-GFP: LD-R-RFP (50:50), and LD-S-GFP: LD-R-RFP (80:20) infected-spleen to SbV as determined from intracellular amastigotes/100 MΦs for calculating EC$_{50}$. **(E.)** DCF-fluorescence quantification of LD-S, LD-R, heat-killed-LD-S, and heat-killed-LD-R infected-MΦs at early hours. Heat-killed LD-R experimental sets at 4 hrs pi significantly fail to generate ROS as compared to LD-R infected-MΦs at 4hrs pi (**\*\***, $P \leq 0.01$). **(F.)** DCF-fluorescence quantification of MΦs infected with LD-S, LD-R, D10, and AmpB-R DD8 **(G.)** DCF-fluorescence quantification of MΦs infected with LD-S or LD-R, or NADPH$^{exh}$-LD-R by $H_2O_2$+6-AN treatment. LD-R ($H_2O_2$+6-AN) significantly failed to generate ROS as compared to LD-R infected-MΦs at 4hrs pi (**\***, $P \leq 0.05$). At 8hrs, LD-R infected- MΦs shows a significant drop in ROS level compared to LD-R ($H_2O_2$+6-AN) (**\*\***, $P \leq 0.01$), which remains unchanged like 4hrs pi. P-value summary is color-coded corresponding to the color of individual experimental sets and all the analysis is performed compared to LD-S infected MΦs at individual time points except for the one in black in S1E and S1G Fig. **\*\*** denotes $P \leq 0.01$, **\*** denotes $P \leq 0.05$, and ns denotes $P > 0.05$. **(H.)** PCA plot showing two separate clustering of LD-S (BHU777 and AG83) demarcated in blue and LD-R (BHU575 and BHU814) demarcated in red. $P > 0.05$ is marked as 'ns' (non-significant) and $P \leq 0.001$ is marked as **\*\*\***. (TIF)

**S2 Fig. Schematic representation of Ferrozine-based iron quantification of PV and cytoplasmic fraction and transcriptomic landscape of LD-S and LD-R-infected-MΦs at 4 hrs and 24 hrs pi as compared to uninfected-MΦs. (A.i.)** Scheme showing PV extraction from LD-S and LD-R infected-MΦs at 4 hrs and 24 hrs pi ensued by quantification of iron following Ferrozine-based colorimetric assay. **(A.ii.)** Heatmap showing iron concentration in cytoplasmic-fraction and PV-fraction of LD-S and LD-R infected-MΦs at 4 hrs and 24 hrs pi. Data are represented as Mean ± SEM. **(B.i.)** Volcano plots of differential gene expression of LD-S (left) and LD-R infected-MΦs (right) versus uninfected control at 4 hrs pi (upper panel) and 24 hrs pi (lower panel). Genes above the significance threshold (adjusted *p-value*<0.05) are marked in green having log$_2$(fold change) >0.6, i.e., upregulated, and red having log$_2$(-fold change) <−0.6, i.e., downregulated, while the rest are marked in grey. **(B.ii.)** Scheme

representing macrophage polarization states (M1 on left and M2 on right) based on the differential gene expression of LD-S (green) and LD-R (red) infected-MΦs at 4 hrs and 24 hrs pi. Score of M1/M2 gene cluster = Sum of respective M1/M2 $\log_2$(fold change) of genes with adjusted *p-value* < 0.05. The determinant score depicting the total polarization outcome is calculated as = Score of M1 gene cluster- Score of M2 gene cluster. At 4 hrs pi, the total score of genes contributing towards M1 polarization in the case of LD-S infection is −0.909 and LD-R infection is 16.237, while the total score of genes contributing towards M2 polarization in the case of LD-S infection is −23.115 and LD-R infection is −10.1189. At 24 hrs pi, the total score of genes contributing towards M1 polarization in the case of LD-S infection is −17.987 and LD-R infection is −52.832, while the total score of genes contributing towards M2 polarization in the case of LD-S infection is −14.385 and LD-R infection is 3.631. **(C.)** Heatmap showing the differential expression pattern of erythophagocytosis-related protein (demarcated in blue), iron-metabolizing and transporter protein (green), and ROS-related proteins (purple) in uninfected-MΦs, LD-S, and LD-R infected-MΦs at 4 hrs and 24 hrs pi (p > 0.5).
(TIF)

**S3 Fig. Localization of Ferroportin and CD71 around LD-S and LD-R PV entailing the iron flux (A.i.)** Confocal images showing the status of CD71 expression (green) in LD-S (upper panel) and LD-R-infected-MΦs (lower panel) at 4 hrs pi. Rab5a (red) demarcates early PV. **(A.ii.a.)** Western blot showing expression of CD71 in the whole cell lysate of uninfected-MΦs, LD-S, and LD-R infected-MΦs at 4 hrs pi. **(A.ii.b.)** Relative intensity of CD71 western blot, i.e., fold change with respect to β-actin where no significant change in the relative intensity of CD71 in UI, LD-S 4hrs pi, and LD-R 4 hrs pi (P > 0.05) is observed. **(B.)** The relative intensity of Ferroportin (Fpn) fold change with respect to β-actin showed significant upregulation of Fpn in both LD-S and LD-R 4 hrs pi (***, P ≤ 0.001) as compared to UI control, with significant upregulation in LD-R 4 hrs pi as compared to LD-S 4hrs pi (***, P ≤ 0.001). **(C.)** Entire panels of Super-resolution images of hepcidin pretreated and hepcidin post-treated sets from Fig 4D.i showing Ferroportin localization. The lowermost panel shows the enlarged view of a portion showing Ferroportin localization with the Rab5a and LD nucleus (small blue dot) **(D.)** Entire panels of live cell confocal images showing iron status by staining with Calcein-AM (green) with lysotracker Red that demarcates PV in hepcidin pretreatment (left 4 panels) and hepcidin post-treatment (right 4 panels) experimental sets (separated by dotted line) elaborated from Fig 4D.ii **(E.)** Entire panels of confocal images of NRAMP1 expression by labeling with anti-NRAMP1 (green) and anti-Rab5a (red) antibodies elaborated from Fig 4E.i.a The dotted area in the merged panel demarcates the MΦ boundary. Yellow arrows show one representative LD-PV with NRAMP1. **(F.)** The relative intensity of NRAMP1, i.e., fold change with respect to β-actin showing significant downregulation of NRAMP1 in both LD-S and LD-R at 4 hrs pi as compared to UI, whereas no significant change is observed in between both these infected experimental conditions. Each densitometry analysis is represented as a bar graph of Mean ± SEM for 3 biological replicates. One representative small nucleus of LD has been marked in (**\***) to show the infected-MΦs. Scale bars indicate 20 μm.
(TIF)

**S4 Fig. Low iron in the cytoplasm in LD-R-infected-MΦs promotes p62/SQSTM1 mediated non-canonically activation of NRF2 (A.i.a.)** Western blot of whole cell lysate showing the expression of p62 in uninfected-MΦs, LD-S, LD-R infected-MΦs at 4 hrs pi and DFO-treated MΦs keeping β-actin as the loading control. **(A.i.b.)** Relative intensity of p62 with respect to β-actin showed a significant upregulation of p62 in both LD-R 4 hrs pi and DFO-treated control as compared to UI (***, P ≤ 0.001). Also, significant upregulation in p62

expression in LD-R 4 hrs pi as compared to LD-S 4hrs pi (****, P ≤ 0.0001) is observed. **(A. ii.)** Confocal images showing the expression pattern of p62 (red) in LD-S and LD-R infected-MΦs at 4 hrs pi. **(B.i.a.)** Western blot of whole cell lysate (left panel) and nuclear fraction (right panel) of uninfected-MΦs, LD-S, and LD-R infected-MΦs at 4 hrs pi showing the expression level of NRF2 keeping β-actin and Histone (H3) as loading control respectively. **(B.i.b.)** The relative intensity of NRF2 from whole cell lysate (left panel) with respect to β-actin showed significant upregulation of NRF2 in both LD-S and LD-R infected MΦs 4 hrs pi (****, P ≤ 0.0001). Also, there is significant upregulation in LD-R 4 hrs pi vs LD-S 4hrs pi (****, P ≤ 0.0001), significant upregulation of nuclear-NRF2 (right panel) in LD-R 4 hrs pi as compared to both UI and LD-S 4hrs pi (***, P ≤ 0.001) is observed. No significant change was observed in the nuclear translocation of NRF2 with respect to uninfected control for LD-S infection. Each densitometry analysis is represented as a bar graph of Mean ± SEM for 3 biological replicates. **(B.ii.a.)** Confocal images showing the localization of NRF2 in LD-S and LD-R infected-MΦs at 4 hrs pi. The left panel shows the RGB-profile plot of NRF2 (red) and DAPI (blue) where the X-axis denotes distance in pixels and the Y-axis denotes intensity. The grey area in the RGB-profile plot shows the colocalized region of NRF2 with the nucleus. **(B. ii.b.)** Stacked bar graphs representing the number of infected macrophages showing nuclear and cytoplasmic NRF2 among LD-S and LD-R-infected-MΦs (N = 30). **(B.iii.)** Super-resolution image showing NRF2 colocalization in the nucleus. The left-most and right-most panel shows 3D rendering of Z-stack images of LD-S and LD-R 4 hrs pi respectively where the yellow arrow denotes a clipped quadrant in Leica Stellaris 5 image processing software to distinguish between superficially-present NRF2 and nuclear-localized NRF2. The middle two panels show the super-resolution Maximum intensity projection of Z-stack images of the respective sets showing the NRF2 (red) expression localization in the nucleus.
(TIF)

**S1 Video. Increased proliferation rate of LD-R-RFP compared to LD-S-GFP** . Videography (20 frames/sec) showing active invasion of LD-S-GFP and LD-R-RFP metacyclic promastigotes in peritoneal macrophages at 4 hrs pi (left panel). The right panel shows LD-R-RFP amastigotes outcompeting LD-S-GFP amastigotes at 24 hrs pi.
(PPTX)

**S2 Video. Differential Ferroportin (green) localization in LD-S and LD-R 24 hrs pi.** Videography (20 frames/sec) showing Ferroportin localized around macrophage surface in LD-S 4hrs pi (left panel) while it surrounds LD (small blue dot represents LD nucleus) in LD-R 4 hrs pi.
(PPTX)

**S3 Video. Differential Furin (red) localization with regard to ADAM10 (magenta) in LD-S and LD-R 24 hrs pi.** Videography (20 frames/sec) showing Furin localization around perinuclear space in LD-S 24hrs pi (left panel) and colocalization with ADAM10 around membrane surface in LD-R 24 hrs pi (right panel) where CD11b (green) is a macrophage membrane raft marker. The right panel of each video represents a snapshot of the 3D Z-stack image with a white arrow demarcating the localization of Furin.
(PPTX)

**S1 Sheet. Excel sheet related to** S2B.ii Fig **showing differential expression of M1 and M2 gene clusters in different experimental conditions (LD-S and LD-R at 4 hrs and 24 hrs pi) from RNAseq analysis.** Genes highlighted in red are significantly downregulated while genes highlighted in green are significantly upregulated.
(XLSX)

**S1 Data. Excel sheet related to data points used to generate the graphs related to this manuscript.**
(XLSX)

## Acknowledgments

We acknowledge DST FIST, Govt. of India, for using the Confocal Microscope of the Biotechnology department, IIT Kharagpur for capturing most of the Confocal images. We acknowledge the Confocal Microscope purchased under the DST-FIST grant conferred on the School of Bioscience, IIT Kharagpur File No. SR/FST/LS-I/2019/595. B.M. acknowledges the SERB International Research Experience (SIRE) fellowship allowing experiments in the Laboratory of Molecular Immunology, Graduate School of Agricultural and Life Sciences, The University of Tokyo, Tokyo, Japan, to perform collaboration experiments. SG acknowledges GATE-MHRD fellowship. We acknowledge the Confocal Microscope of the Leica Stellaris 5 Dmi8 demonstration unit for capturing images of Fig 3. We would like to acknowledge the Central Research Facility of the School of Medical Science and Technology, IIT Kharagpur, India for providing access to instruments such as Flow Cytometer (BD Biosciences) and ERBA H560 Fully Automated 5-part Hematology Analyser. We would like to acknowledge Debolina Manna, Ph.D. scholar in IDI lab, SMST, IITKGP for generating transgenic LD-S-GFP, LD-R-GFP, and LD-R-RFP lines.

## Author contributions

**Conceptualization:** Souradeepa Ghosh, Budhaditya Mukherjee.

**Formal analysis:** Souradeepa Ghosh, Krishna Vamshi Chigicherla, Shirin Dasgupta.

**Funding acquisition:** Shirin Dasgupta, Yasuyuki Goto, Budhaditya Mukherjee.

**Investigation:** Souradeepa Ghosh, Krishna Vamshi Chigicherla, Shirin Dasgupta, Yasuyuki Goto, Budhaditya Mukherjee.

**Methodology:** Souradeepa Ghosh, Yasuyuki Goto, Budhaditya Mukherjee.

**Project administration:** Yasuyuki Goto, Budhaditya Mukherjee.

**Supervision:** Budhaditya Mukherjee.

**Visualization:** Souradeepa Ghosh, Krishna Vamshi Chigicherla, Budhaditya Mukherjee.

**Writing – original draft:** Souradeepa Ghosh, Budhaditya Mukherjee.

**Writing – review & editing:** Souradeepa Ghosh, Krishna Vamshi Chigicherla, Shirin Dasgupta, Yasuyuki Goto, Budhaditya Mukherjee.

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
