## [Decision Letter · Decision Letter 0]

29 Jul 2024

Dear Dr. Mukherjee,

Thank you very much for submitting your manuscript "Oxidative stress-driven enhanced iron production and scavenging through Ferroportin reorientation worsens anemia in antimony-resistant Leishmania donovani infection." (PPATHOGENS-D-24-00843) for consideration at PLOS Pathogens. As with all papers peer reviewed by the journal, your manuscript was reviewed by members of the editorial board and by several independent peer reviewers. Based on the reports, we regret to inform you that we will not be pursuing this manuscript for publication at PLOS Pathogens.

The reviewers did not support publication of the manuscript. Reviewer #1 in particular raised a number of methodological issues that raise some important concerns about the robustness of the data.

The reviews are attached below this email, and we hope you will find them helpful if you decide to revise the manuscript for submission elsewhere. We are sorry that we cannot be more positive on this occasion. We very much appreciate your wish to present your work in one of PLOS's Open Access publications.

Thank you for your support, and we hope that you will consider PLOS Pathogens for other submissions in the future.

Sincerely,

Jeffrey Dvorin

Section Editor

PLOS Pathogens

Michael Malim

Editor-in-Chief

PLOS Pathogens

orcid.org/0000-0002-7699-2064

Reviewer's Responses to Questions

**Part I - Summary**

Reviewer #1: The manuscript contains significant flaws that undermine the validity and reliability of its conclusions. The main issues are outlined below.

Given these critical issues, the manuscript's conclusions are not substantiated by the data presented. The inconsistencies, lack of essential data, and inadequate methodological details significantly diminish the credibility and scientific contribution of this manuscript. Therefore, the manuscript is recommended for rejection.

Reviewer #2: The manuscript entitled “Oxidative stress-driven enhanced iron production and scavenging through Ferroportin reorientation worsens anemia in antimony-resistant Leishmania donovani Infection” has investigated possible mechanisms that aid the persistence of Antimony resistant L. donovani parasites, in the Indian subcontinent even after the cessation of antimony treatment in the clinics. These parasites persist better in the host than antimony-sensitive ones. They show that resistant parasites to antimonials scavenge iron more efficiently in the parasitophorous vacuole, than sensitive ones and hence can replicate better as amastigotes.

With the use of an antioxidant NAC, they show ROS are required for the survival of the resistant parasites and this was associated with upregulation of OH-1, but this increase was associated with a downmodulation of cytoplasmic Fe+2, contrary to what it was believed. This was associated with an increase of iron, at 4 hr pi of LD-R-Parasitophorous Vacuole. The latter was associated with targeted redistribution of the MФ membrane iron-exporter, Ferroportin, around LD-R-PV, where the authors say that this causes a downmodulation of cytoplasmic iron and increases the activation of p62/SQSTM1. This is known to degrade the inhibitor of Nrf1 Keap1, facilitating the translocation of Nrf1 to the nucleus, a transcription factor known to initiate the antioxidant response. This iron insufficiency results in increased erythrophagocytosis, via SIRPα degradation, mediated by furin and ADAM10. The authors suggest that erythrophagocytosis is the cause of anaemia, observed in patients infected with LD-R parasites, as it is the case for infected mice with LD-R parasites. This manuscript is novel and has a lot of different experimental procedures and explains contradicting observations in macrophage-Leishmania interactions, like the activation of ROS for the benefit of the parasite. Ηοwever the presentation is long and complex and the authors need to reassure the readers about the quality of some experiments (described in major comments). Overall, the positive elements of this manuscript make it worth for publication, if the major comments are addressed.

**Part II – Major Issues: Key Experiments Required for Acceptance**

Reviewer #1: 1) Lack of Supporting Data:

Organ-Parasite Load: The authors conclude about "organ-parasite" load in the 1st Results section without presenting any data on data. This omission makes the conclusion unsupported.

Organ Weight: Similarly, the discussion on significant differences in organ weight is not backed by any data on organ weight, leading to unsupported claims.

2) Inconsistent Methodology:

In Vitro vs. In Vivo Comparisons: For in vitro infection, a GFP susceptible strain and a RFP resistant strain are compared, but for in vivo experiments, a GFP susceptible strain is compared with a non-fluorescent resistant strain. The methodology should be consistent, and the same tagged resistant strain should be used in both in vitro and in vivo experiments to track both strains in microscopy and citometry experiments.

3) Missing Data and Results:

No results are presented for the description on lines 156-158.

The data does not show differences among Groups 2, 3, and 4, contrary to the discussion on lines 143-148.

Essential data on parasite proliferation after 48 and 72 hours of in vitro infection and in vivo organ-parasite load, are missing.

4) Poor Image Quality and Inadequate Controls:

Confocal images in Fig 1C.ii and FigS1D.i are of poor resolution and do not support any conclusions.

Figure 2A lacks scale bars and controls, making it difficult to interpret the results accurately.

5)Unsupported Conclusions:

ROS Levels: Conclusions on ROS levels differences between LD-S and LD-R strains are not supported by the presented data and statistical analysis (lines 168-171).

Ferritin Levels: Conclusions on Ferritin levels are not supported by the densitometry analysis of the western blot in Fig. 3B. Quantification of HO-1 levels and biological replicates for the western blot analysis are also missing.

Luciferase Activity: The conclusion that luciferase activity modulation is exclusive to LD-R is not accurate, as activity modulation is also observed in LD-S.

6) Inadequate Statistical Analysis:

Statistical analyses for several conclusions, including anemia-related findings, are not presented. Specifically, the HCT differences between LD-S and LD-R are not statistically significant (t test done by myself).

7) Methodological Concerns:

The percentages of metacyclics shown in Fig 1Ai, Aii and Fig S1A are before or after sorting? Most published works report percentages below 40% before purification.

Calcein is not a reliable method for LIP quantification in this context (doi: 10.1042/BJ20061840, among others).

Many western blot-related conclusions are quantitative, yet the manuscript does not report how these quantifications were performed or the number of replicates.

Reviewer #2: A lot of “unwanted information” which does not contribute to the final conclusion (ie increased iron pool in PV with lower amounts in cytoplasm which contributes to increase in erythrophagocytosis). I would present the result with the low ironOH-1 upregulation in the supplemental material. I would like to see a simplification and more results in the supplemental figures. For some experiments like the transfection of RAW macrophages with dual reporter luciferase, I would like the authors to comment as it is in general established the macrophages cannot be easily transfected with DNA. Also, as the authors present transcriptomics results provided as a service, but there is no clear mention of the quality of these data in the main MS (number of replicas etc). Also, many of these factors that are upregulated or downmodulated, apart from the Iron pathway affect the whole macrophage function (ie OH-1 causes a M2 phenotype in macrophages and hence allows the growth of intracellular parasites). In the discussion, it would be great to see some description between these results and the macrophage programs (inflammatory M1, anti-inflammatory M2).

**Part III – Minor Issues: Editorial and Data Presentation Modifications**

Reviewer #1: (No Response)

Reviewer #2: Some typos and grammatical errors. Ie line 407 intracellular proliferation (requires a full stop). I would also like to see an abbreviation list.Also in the results section it is not always easy to understand if some experiments are performed in promastigites, amastigotes or macrophages

PLOS authors have the option to publish the peer review history of their article (what does this mean? ). If published, this will include your full peer review and any attached files.

**Do you want your identity to be public for this peer review?** For information about this choice, including consent withdrawal, please see our Privacy Policy .

Reviewer #1: No

Reviewer #2: No

---

## [Editor Report · Decision Letter 1]

12 Aug 2024

Dear Dr. Mukherjee,

Thank you very much for submitting your manuscript "Oxidative stress-driven enhanced iron production and scavenging through Ferroportin reorientation worsens anemia in antimony-resistant Leishmania donovani infection." for consideration at PLOS Pathogens. As with all papers reviewed by the journal, your manuscript was reviewed by members of the editorial board and by several independent reviewers. In light of the reviews (below this email), we would like to invite the resubmission of a significantly-revised version that takes into account the reviewers' comments.

We cannot make any decision about publication until we have seen the revised manuscript and your response to the reviewers' comments. Your revised manuscript is also likely to be sent to reviewers for further evaluation.

Sincerely,

Jeffrey D Dvorin, MD, PhD

Section Editor

PLOS Pathogens

Jeffrey Dvorin

Section Editor

PLOS Pathogens

Michael Malim

Editor-in-Chief

PLOS Pathogens

orcid.org/0000-0002-7699-2064

Figure Files:

While revising your submission, please upload your figure files to the Preflight Analysis and Conversion Engine (PACE) digital diagnostic tool, https://pacev2.apexcovantage.com. PACE helps ensure that figures meet PLOS requirements. To use PACE, you must first register as a user. Then, login and navigate to the UPLOAD tab, where you will find detailed instructions on how to use the tool. If you encounter any issues or have any questions when using PACE, please email us at

Data Requirements:

Please note that, as a condition of publication, PLOS' data policy requires that you make available all data used to draw the conclusions outlined in your manuscript. Data must be deposited in an appropriate repository, included within the body of the manuscript, or uploaded as supporting information. This includes all numerical values that were used to generate graphs, histograms etc.. For an example see here on PLOS Biology: http://www.plosbiology.org/article/info%3Adoi%2F10.1371%2Fjournal.pbio.1001908#s5 .
---

## [Decision Letter · Decision Letter 2]

23 Dec 2024

Dear Dr. Mukherjee,

We are pleased to inform you that your manuscript 'Oxidative stress-driven enhanced iron production and scavenging through Ferroportin reorientation worsens anemia in antimony-resistant Leishmania donovani infection' has been provisionally accepted for publication in PLOS Pathogens.

While not absolutely required, it would be helpful to make some edits in response to the reviewer 1 comment about reducing redundancy to improve clarity. 

Best regards,

Jeffrey D Dvorin, MD, PhD

Section Editor

PLOS Pathogens

Jeffrey Dvorin

Section Editor

PLOS Pathogens

Sumita Bhaduri-McIntosh

Editor-in-Chief

PLOS Pathogens

orcid.org/0000-0003-2946-9497

Michael Malim

Editor-in-Chief

PLOS Pathogens

orcid.org/0000-0002-7699-2064

Reviewer Comments (if any, and for reference):

Reviewer's Responses to Questions

**Part I - Summary**

Reviewer #1: The manuscript titled "Oxidative stress-driven enhanced iron production and scavenging through Ferroportin reorientation worsens anemia in antimony-resistant Leishmania donovani infection" investigates the mechanisms by which L. donovani develops resistance to pentavalent antimonials, focusing on the parasite’s ability to manipulate host iron homeostasis and ROS production. The study highlights how antimony-resistant L. donovani (LD-R) adapts to oxidative stress and enhances iron acquisition through a reorientation of Ferroportin on the macrophage surface. The manuscript presents novel insights into the metabolic shifts, which enable LD-R to thrive in iron-limited environments. These findings offer valuable mechanistic insights that may also inform our understanding of drug resistance in other intracellular pathogens.

Upon reviewing the revised manuscript, I appreciate the authors' efforts in addressing the major concerns raised in my initial review. The manuscript has been significantly strengthened, particularly in terms of data presentation and consistency in methodology. The new data comparing the GFP-susceptible strain and RFP-resistant strain provide clearer results. Additionally, the statistical analysis of the HCT data has been improved with a larger sample size, which strengthens the reliability of the conclusions.

While the authors have successfully addressed most of the major issues, a few minor revisions are still needed. Specifically, the authors should focus on trimming unnecessary content and refining the presentation of data. Therefore, I recommend accepting the manuscript with minor revisions.

Reviewer #2: The authors of the manuscript "Oxidative stress-driven enhanced iron production and scavenging through Ferroportin

reorientation worsens anemia in antimony-resistant Leishmania donovani infection" have replied to most comments adequately. However some small issues could be improved.

**Part II – Major Issues: Key Experiments Required for Acceptance**

Reviewer #1: (No Response)

Reviewer #2: I would like to see further validation of the "omics" data with 2 replicates. Ie I would like the authors "to select more gene transcripts" and assess the expression and compare it with the data provided.

**Part III – Minor Issues: Editorial and Data Presentation Modifications**

Reviewer #1: - Figures:

Although there have been improvements, some figures, particularly Fig.2E.i and Fig.2E.ii, still suffer from low resolution and are too small. When zooming in, the resolution does not allow for clear reading of the axis labels. These adjustments are necessary to improve interpretability and to better support the manuscript’s conclusions.

- Excessive Length and Unnecessary Information:

A significant minor issue is that the manuscript contains a considerable amount of information that does not directly contribute to the final conclusions. Several sections, particularly in the introduction and discussion, are lengthy and could be streamlined. The manuscript would benefit from a more concise presentation of the data and a tighter focus on the core message. Currently, the text is somewhat difficult to follow due to its length and the inclusion of tangential details. Reducing redundancy and removing unnecessary explanations would improve clarity and readability.

Reviewer #2: (No Response)

PLOS authors have the option to publish the peer review history of their article (what does this mean? ). If published, this will include your full peer review and any attached files.

**Do you want your identity to be public for this peer review?** For information about this choice, including consent withdrawal, please see our Privacy Policy .

Reviewer #1: No

Reviewer #2: No

---

## [Editor Report · Acceptance letter]

Dear Dr. Mukherjee,

We are delighted to inform you that your manuscript, "Oxidative stress-driven enhanced iron production and scavenging through Ferroportin reorientation worsens anemia in antimony-resistant Leishmania donovani infection," has been formally accepted for publication in PLOS Pathogens.

Best regards,

Sumita Bhaduri-McIntosh

Editor-in-Chief

PLOS Pathogens

orcid.org/0000-0003-2946-9497

Michael Malim

Editor-in-Chief

PLOS Pathogens

orcid.org/0000-0002-7699-2064